# R-Diverse: Mitigating Diversity Illusion in Self-Play LLM Training

**Gengsheng Li** [* 1 2] **Jinghan He** [* 1 2] **Shijie Wang** [1 2] **Ruiqi Liu** [2] **Renrui Zhang** [2] **Zijun Yao** [3] **Junfeng Fang** [4] **Haiyun Guo** [1 2] **Dan Zhang** [4] **Jinqiao Wang** [1 2 5]

## Abstract

Self-play bootstraps LLM reasoning through an iterative Challenger–Solver loop: the Challenger is trained to generate questions that target the Solver's capabilities, and the Solver is optimized on the generated data to expand its reasoning skills. However, existing frameworks like R-Zero often exhibit non-sustained improvement, where early gains degrade as self-play continues. We identify a key failure mode, **Diversity Illusion**, where the Solver's training signals appear diverse yet collapse into recurring underlying patterns. It manifests as (1) *Local Diversity Illusion*, where diversity is enforced only within-batch, inducing cross-iteration mode cycling; and (2) *Surface Diversity Illusion*, where questions vary superficially but require near-identical reasoning skills. To mitigate them, we propose R-Diverse with two aligned innovations: **Memory-Augmented Penalty (MAP)**, which uses a persistent memory bank to discourage recycling across iterations, and **Skill-Aware Measurement (SAM)**, which evaluates diversity by the reasoning skills exercised rather than surface variation of questions. Across 10 math and general reasoning benchmarks, R-Diverse sustains gains over more iterations and consistently outperforms prior self-play methods.

## 1. Introduction

Self-play has enabled strong performance gains by letting an agent improve through competition with itself, most notably in game-playing AI (Silver et al., 2017; 2018; Vinyals et al., 2019). Recently, this paradigm has been adapted to large language models (LLMs) as an iterative Challenger–Solver loop for self-improvement and alignment (Chen et al., 2024; Yuan et al., 2024; Zhang et al., 2024b), and has become a promising route for training reasoning-oriented LLMs (Zhang et al., 2024a; Zhao et al., 2025; Wang et al., 2025a; Liu et al., 2025; Huang et al., 2025; Guo et al., 2025). In this setting, the Challenger is optimized to generate questions that expand the Solver's capabilities, and the Solver is then trained on these generated questions as self-produced training signals; repeating this alternation drives their co-evolution. Despite this promise, current self-play frameworks for reasoning often yield non-sustained gains: performance improves early but plateaus or degrades after a few iterations (Yu et al., 2025; Kuba et al., 2025; Shumailov et al., 2024). This brittleness remains a key obstacle to reliable self-evolving LLM training.

To understand this challenge, we diagnose a key failure mode, **Diversity Illusion**. That is the Challenger can satisfy diversity constraints in appearance while the training signals used to update the Solver progressively concentrate on recurring underlying patterns, resulting in limited reasoning skills exposure. Concretely, most self-play frameworks (Huang et al., 2025; He et al., 2025; Yu et al., 2025) impose a repetition penalty that discourages surface-level similarity among generated questions within the current batch. In contrast, we monitor repetition from two complementary perspectives: *Cross-Iteration Repetition*, which captures historical recycling by measuring how often newly generated questions resemble those from previous iterations, and *Intra-Iteration Repetition*, which captures within-iteration homogenization by measuring how much the questions within the same iteration collapse to similar underlying requirements. As shown in Figure 1(a), while the Repetition Penalty consistently decreases, both Cross-Iteration and Intra-Iteration Repetition steadily rise. It reveals a divergence between what the objective penalizes and what the Solver is actually trained on. This divergence exposes two subtypes of diversity illusion (Figure 1(b)): (1) **Local Diversity Illusion**, where enforcing diversity only within-batch without historical memory induces cross-iteration mode cycling; and (2) **Surface Diversity Illusion**, where questions vary superficially yet require near-identical reasoning skills even within

---

[*]Equal contribution   [1]Foundation Model Research Center, Institute of Automation, Chinese Academy of Sciences [2]School of Artificial Intelligence, University of Chinese Academy of Sciences [3]Tsinghua University [4]National University of Singapore [5]Wuhan AI Research. Correspondence to: Haiyun Guo <haiyun.guo@nlpr.ia.ac.cn>, Dan Zhang <zhang-dan25@nus.edu.sg>.

*Proceedings of the 43rd International Conference on Machine Learning*, Seoul, South Korea. PMLR 306, 2026. Copyright 2026 by the author(s).

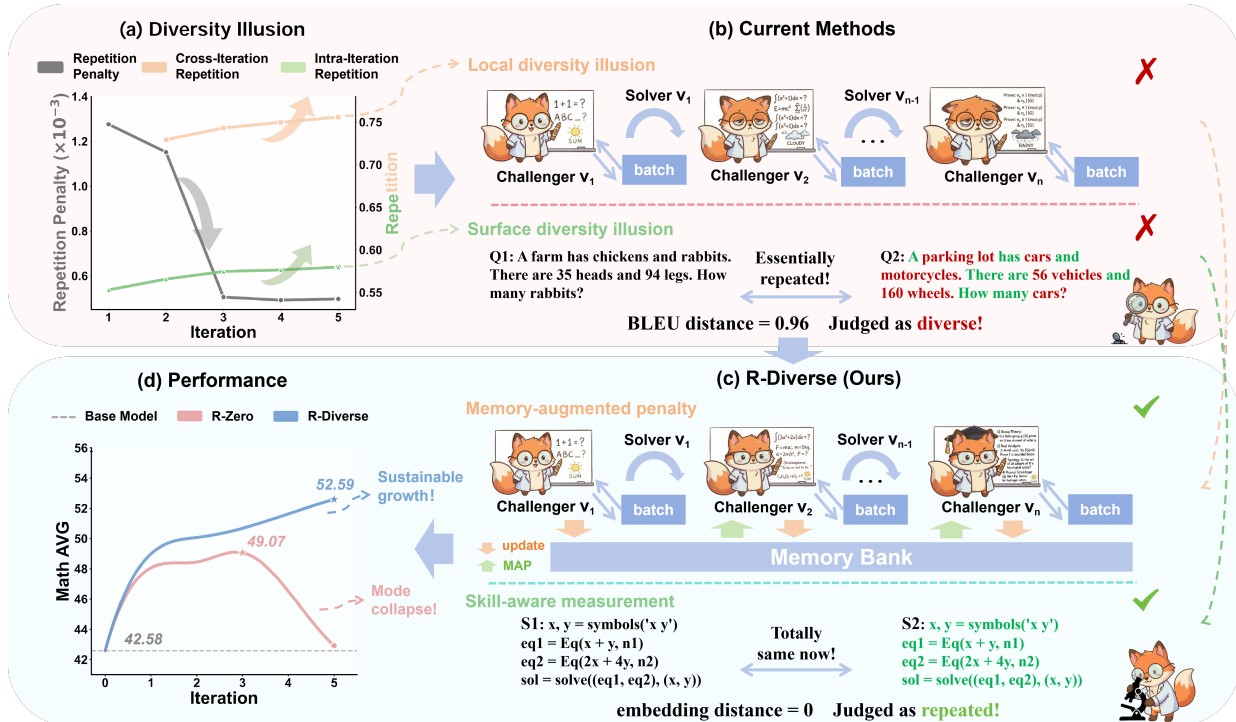

*Figure 1.* Overview of Diversity Illusion and the R-Diverse framework. **(a)** Despite a decreasing repetition penalty, cross-iteration and intra-iteration repetition increase, revealing a mismatch between what is penalized and what the Solver is trained on. **(b)** Existing methods exhibit *Local Diversity Illusion* and *Surface Diversity Illusion*. **(c)** R-Diverse resolves these failures with MAP to enforce global, history-aware exploration and SAM to identify repetitions at the level of underlying reasoning skills. **(d)** Consequently, R-Diverse sustains improvement over five iterations (52.59), avoiding the collapse observed in R-Zero.

the same iteration.

Building on this diagnosis, we propose **R-Diverse** (Figure 1(c)), a self-play framework that promotes both cross-iteration and skill-aware diversity in the training signals for more sustainable evolution. R-Diverse introduces two aligned innovations. First, **Memory-Augmented Penalty (MAP)** addresses the *Local Diversity Illusion* by equipping the Challenger with a persistent memory bank and penalizing similarity to previously generated questions, thereby discouraging cross-iteration mode cycling. We additionally use the memory bank for experience replay, mixing a small ratio of historical samples into each iteration to temper distribution shift and maintain competence on previously learned skills. Second, **Skill-Aware Measurement (SAM)** targets the *Surface Diversity Illusion* by redefining diversity in terms of the reasoning skills exercised by the Solver, rather than surface variation in question statements. In our implementation, we operationalize this skill-aware assessment in two steps: (i) a representation abstraction step that maps each question to a canonical solver-level program representing its solution procedure; and (ii) a similarity computation step that embeds these solutions using an off-the-shelf encoder and measures repetition via embedding similarity, so that the diversity measure reflects differences

in the reasoning skills being exercised.

We evaluate R-Diverse on the Qwen3 family across seven mathematical and three general reasoning benchmarks. As shown in Figure 1(d), R-Diverse sustains improvement over five evolution iterations, while the prior self-play baseline R-Zero typically plateaus or degrades after three iterations, trending back toward base-model performance. R-Diverse consistently outperforms previous self-play methods across domains and model scales, providing a more reliable recipe for iterative self-play training. Code is available at https://github.com/Gengsheng-Li/R-Diverse.

Our contributions are threefold:

- We diagnose Diversity Illusion as a key failure mode in self-play reasoning, and decompose it into Local Diversity Illusion and Surface Diversity Illusion.

- We propose R-Diverse, a self-play framework with MAP to enforce global diversity, and SAM to evaluate diversity by reasoning skills exercised rather than surface variation of questions.

- Extensive experiments demonstrate that R-Diverse enables more sustainable self-improvement and consistently outperforms prior methods.

## 2. Preliminaries

Our work builds upon the self-play paradigm for training reasoning LLMs. In this section, we formalize this framework using R-Zero (Huang et al., 2025) as a representative instance. We first detail the co-evolutionary objectives of both the Challenger and the Solver, and then analyze the critical limitations in existing diversity mechanisms.

### 2.1. Self-Play Framework for Reasoning LLMs

We consider a standard self-play setup involving two agents initialized from a base model $M_0$: a Challenger $\mathcal{C}_\theta$ that generates questions $q$, and a Solver $\mathcal{S}_\phi$ that generates solution paths $a$. The training process alternates between two phases across iterations $t = 1, \ldots, T$:

**Challenger Phase.** The Solver $\mathcal{S}_{\phi_{t-1}}$ is frozen. The Challenger $\mathcal{C}_{\theta_t}$ is trained to generate questions that lie in the Solver's "zone of proximal development", where problems are challenging but solvable. Its optimization objective maximizes a composite reward $R_{\mathcal{C}}$:

$$R_{\mathcal{C}}(q) = R_{\text{uncertainty}}(q) - \lambda \cdot P_{\text{rep}}(q), \qquad (1)$$

where the uncertainty reward $R_{\text{uncertainty}}$ encourages appropriate difficulty. For each question $q$ generated by the Challenger, the Solver produces $K$ responses. We extract final answers from these responses and partition them into groups $\mathbb{G} = \{G_1, \ldots, G_k\}$ based on answer equivalence. The consistency score $s(q)$ is defined as the ratio of the largest group size to $K$:

$$s(q) = \frac{\max_{G \in \mathbb{G}} |G|}{K}, \quad s(q) \in [0, 1]. \qquad (2)$$

The reward $R_{\text{uncertainty}}$ peaks when $s(q) = 0.5$, indicating that the question lies at the boundary of the Solver's capability:

$$R_{\text{uncertainty}}(q) = \min(s(q), 1 - s(q)). \qquad (3)$$

The repetition penalty $P_{\text{rep}}$ discourages repetition within the current batch $\mathcal{B}$. R-Zero implements this via agglomerative clustering based on pairwise BLEU (Papineni et al., 2002) distance $d_{ij} = 1 - \text{BLEU}(q_i, q_j)$. For a question $q_i$ belonging to cluster $C_i$, the penalty is the cluster's relative size:

$$P_{\text{rep}}(q_i, \mathcal{B}) = \frac{|C_i|}{|\mathcal{B}|}. \qquad (4)$$

**Solver Phase.** The Challenger $\mathcal{C}_{\theta_t}$ is frozen and generates a curriculum of new questions. The Solver $\mathcal{S}_{\phi_t}$ is trained on these questions using pseudo-labels derived from its own high-confidence outputs (e.g., via majority voting). Its reward $R_{\mathcal{S}}$ is a simple binary signal:

$$R_{\mathcal{S}}(a, y) = \mathbb{I}(\text{answer}(a) = \hat{y}), \qquad (5)$$

where $\hat{y}$ is the pseudo-label. This encourages the Solver to master the curriculum proposed by the Challenger.

Both agents are optimized using Group Relative Policy Optimization (GRPO; Shao et al., 2024), which eliminates the need for a critic model by estimating advantages from group-normalized rewards across multiple sampled responses.

### 2.2. Limitations: The Roots of Diversity Illusion

While the R-Zero framework ostensibly provides a solid foundation, its repetition penalty mechanism suffers from two structural flaws that lead to the Diversity Illusion:

**Local Scope.** Since the Challenger is blind to previously generated questions, the penalty $P_{\text{rep}}$ is conditioned solely on the current batch $\mathcal{B}$. Theoretically, minimizing $R_{\mathcal{C}}$ only encourages generating questions that are distinct within $\mathcal{B}$ but potentially similar to high-reward questions from previous iterations. This limitation leads to the **Local Diversity Illusion**, where global repetition rises despite local constraints.

**Surface Metric.** Prior self-play frameworks typically penalize repetition using *question-level* surface metrics (e.g., BLEU), which quantify diversity in the token space $\mathcal{X}$. However, what matters for self-play training is the diversity of *training signals for the Solver*—namely the underlying reasoning skills required to solve the questions. Let $\mathcal{Z}$ denote this skill space, and let $f : \mathcal{X} \to \mathcal{Z}$ map a question statement to its required reasoning skills. This mapping is generally many-to-one: many superficially different questions can induce near-identical reasoning skills. Consequently, the Challenger may produce $q_i, q_j$ with low $\text{BLEU}(q_i, q_j)$ yet $f(q_i) \approx f(q_j)$. We refer to this mismatch as the **Surface Diversity Illusion**.

These limitations motivate a framework that (i) enforces diversity *across iterations* rather than within a local batch, and (ii) measures diversity at the level of *reasoning skills* exercised rather than surface question variation, leading to our proposed **R-Diverse**.

## 3. Method

As illustrated in Figure 2, R-Diverse introduces two complementary mechanisms: (1) MAP, which leverages a memory bank to enforce *cross-iteration* exploration by penalizing similarity to previously generated questions; and (2) SAM, which promotes *skill-aware* diversity by shifting comparison from surface question forms to underlying reasoning skills.

### 3.1. Memory-Augmented Penalty

MAP addresses the Local Diversity Illusion by equipping the Challenger with a persistent memory bank $\mathcal{M}$, expand-

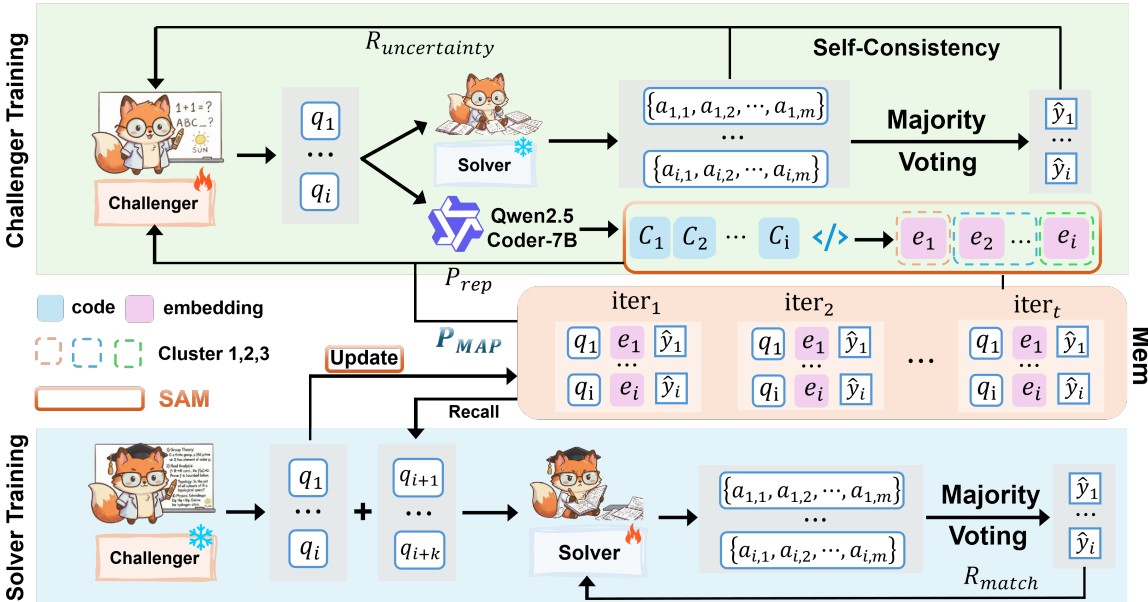

*Figure 2.* The R-Diverse framework. **Top (Challenger training)**: the Challenger proposes questions $\{q_i\}$ to maximize the uncertainty reward $R_{\text{uncertainty}}$. For each $q_i$, the Solver produces multiple solutions $\{a_{i,1}, a_{i,2}, \ldots, a_{i,m}\}$ and a pseudo-label $\hat{y}_i$ via majority voting. SAM maps $q_i$ to canonical solver code $c_i$ (via Qwen2.5-Coder-7B) and a semantic embedding $e_i$, which are used to compute both the within-iteration repetition penalty $P_{\text{rep}}$ and the memory-augmented penalty $P_{\text{MAP}}$. **Middle (Memory update)**: the memory bank $\mathcal{M}$ stores historical tuples $(q_i, e_i, \hat{y}_i)$ across iterations. **Bottom (Solver training)**: to mitigate distribution shift, the Solver is trained on current questions augmented with samples recalled from $\mathcal{M}$, using a matching reward $R_{\text{match}}$.

ing its visibility from the local batch to the entire evolutionary history.

### 3.1.1. GLOBAL MEMORY BANK CONSTRUCTION

We maintain a memory bank $\mathcal{M}$ containing the embedding vectors of all valid questions generated in previous iterations. Let $\phi(q)$ be a semantic embedding function. While we formally instantiate $\phi$ using our SAM in Section 3.2, we first describe how MAP utilizes this representation to enforce global diversity. Let $\mathcal{Q}_t$ denote the set of questions generated by the Challenger at iteration $t$. At the end of iteration $t$, we update the memory:

$$\mathcal{M}_{t+1} \leftarrow \mathcal{M}_t \cup \{\phi(q) \mid q \in \mathcal{Q}_t, \text{valid}(q)\}, \quad (6)$$

where valid($q$) denotes questions that pass the uncertainty filter (e.g., $0.3 \leq s(q) \leq 0.8$), ensuring we only store high-quality training data.

### 3.1.2. DUAL-PERSPECTIVE PENALTY

To effectively penalize the Challenger, we introduce the MAP penalty $P_{\text{MAP}}(q, \mathcal{M})$, composed of two distinct terms that target specific failure modes:

**Max-Similarity Penalty.** We first penalize the maximum similarity to any single historical sample to prevent direct repetition:

$$P_{\text{max}}(q, \mathcal{M}) = \max_{e \in \mathcal{M}} \cos(\phi(q), e). \quad (7)$$

This ensures the new question is distinct from every specific past instance. However, $P_{\text{max}}$ alone is insufficient: a question can be distinct from any single predecessor yet still fall within a dense region of previously explored topics, failing to expand the semantic frontier.

**Mean-Similarity Penalty.** To complement $P_{\text{max}}$ and drive exploration into a sparse region, we penalize the average similarity to the entire memory bank:

$$P_{\text{mean}}(q, \mathcal{M}) = \frac{1}{|\mathcal{M}|} \sum_{e \in \mathcal{M}} \cos(\phi(q), e). \quad (8)$$

While $P_{\text{max}}$ prevents point-to-point collision, $P_{\text{mean}}$ prevents point-to-region collision. It acts as a global repulsive force that pushes the Challenger away from the "center of gravity" of historical question distributions, encouraging the exploration of novel, under-represented question spaces.

The final $P_{\text{MAP}}$ is a weighted combination, activated only when similarity exceeds a tolerance threshold $\tau$:

$$\begin{aligned} P_{\text{MAP}}(q, \mathcal{M}) = \gamma \cdot [P_{\text{max}}(q, \mathcal{M}) - \tau_{\text{max}}]_+ \\ + (1 - \gamma) \cdot [P_{\text{mean}}(q, \mathcal{M}) - \tau_{\text{mean}}]_+, \end{aligned} \quad (9)$$

where $[\cdot]_+ = \max(0, \cdot)$. We set $\gamma = 0.5$ to balance pointwise novelty and distributional exploration.

**Stabilization via Experience Replay.** The effectiveness of MAP in driving global exploration inevitably results in a continuously shifting question distribution. To ensure the Solver robustly masters this expanding curriculum without forgetting learned skills, we incorporate a memory replay mechanism during Solver training. Specifically, at each iteration $t$, we augment the current iteration data $\mathcal{D}_t$ by sampling historical high-quality question-answer pairs $\mathcal{D}_{\text{history}}$ from $\mathcal{M}_{t-1}$, ensuring these historical samples constitute a target ratio $\rho$ of the final training set. This strategy ensures the Solver maintains competence on diverse problem types even as the Challenger's distribution evolves.

### 3.2. Skill-Aware Measurement

We instantiate SAM via two steps. (i) Representation Abstraction: we map a natural-language question to a canonical solver-level program to capture the underlying solution procedure while filtering out narrative phrasing. (ii) Embedding-based Similarity Computation: we embed the program using an off-the-shelf code encoder and compute similarity in embedding space, providing a robust diversity measure that is less sensitive to superficial code edits yet responsive to meaningful procedural differences.

**Representation Abstraction.** We use Python solver code as a solver-level bottleneck that strips away linguistic variation and retains the core reasoning procedure. Concretely, we prompt a code generation model, Qwen2.5-Coder-7B (Hui et al., 2024), to translate a natural-language question $q$ into a canonical function $\text{Code}(q)$ that reflects the underlying relations rather than narrative details. To encourage canonicalization, we set temperature to 0 and apply two constraints: (1) *Procedure Extraction*, requiring the code to express the underlying mathematical relations; and (2) *Symbolic Anonymization*, replacing concrete entities (e.g., "apples") with generic variables (e.g., $x, y$).

As illustrated in Figure 1(b, c), this mapping sends textually distinct but procedurally similar problems (e.g., "chickens/rabbits" vs. "cars/wheels") to highly similar code structures. Even when the generated code is imperfect, it typically preserves the dependency structure and operation patterns, serving as a practical proxy for the reasoning skills exercised.

**Embedding-based Similarity Computation.** Residual surface variation may remain at the code level (e.g., constants or minor syntactic choices), making string-level matching brittle. We therefore encode $\text{Code}(q)$ into a continuous vector using an off-the-shelf code encoder (Jina-Code-Embeddings-1.5B (Kryvosheieva et al., 2025)):

$$\phi_{\text{SAM}}(q) = \text{Encoder}(\text{Code}(q)). \qquad (10)$$

We compute similarity by cosine distance in this embedding space, which is less sensitive to superficial code edits yet responsive to meaningful differences in underlying procedures.

**Integration.** We use $\phi_{\text{SAM}}$ as the similarity function in both the repetition penalty $P_{\text{rep}}$ and the memory-augmented penalty $P_{\text{MAP}}$, so that R-Diverse consistently enforces diversity in terms of the reasoning skills exercised rather than surface variation in questions.

### 3.3. R-Diverse Training

We integrate the proposed MAP and SAM components into the R-Zero framework, yielding the following training objectives:

**Challenger Training.** The Challenger $\mathcal{C}_\theta$ maximizes a composite reward that balances difficulty with both within-iteration and cross-iteration novelty, measured by a skill-aware similarity function:

$$
\begin{aligned}
R_{\text{total}}(q) = R_{\text{uncertainty}}(q) - \alpha P_{\text{rep}}(q, \mathcal{B}; \phi_{\text{SAM}}) \\
- \beta P_{\text{MAP}}(q, \mathcal{M}; \phi_{\text{SAM}}).
\end{aligned}
\qquad (11)
$$

Unlike R-Zero which relies on lexical BLEU, we unify both $P_{\text{rep}}$ and our $P_{\text{MAP}}$ under the skill-aware $\phi_{\text{SAM}}$.

**Solver Training.** The Solver $\mathcal{S}_\phi$ objective follows the standard formulation in Eq. 5, optimizing performance on the evolving curriculum proposed by the Challenger (with optional memory replay as described in Sec. 3.1).

## 4. Experiments

### 4.1. Experimental Setup

**Implementation.** We implement R-Diverse on top of the EasyR1 codebase (Zheng et al., 2025) and use Qwen3-4B-Base and Qwen3-8B-Base (Yang et al., 2025) as base models. Each evolution iteration consists of 5 training steps for the Challenger and 15 training steps for the Solver. We set $\alpha = 1.0$, $\beta = 1.0$, $\gamma = 0.5$, $\tau_{\max} = 0.5$, $\tau_{\text{mean}} = 0.25$, and use a memory replay ratio $\rho = 0.3$. Full hyperparameters and prompts are provided in App. A.

**Comparison Methods.** We compare against representative self-play methods including R-Zero (Huang et al., 2025), Absolute Zero (Zhao et al., 2025), SPIRAL (Liu et al., 2025), and Socratic-Zero (Wang et al., 2025a), as well as the base models. Among these, only R-Zero adopts a discrete iteration-based evolution comparable to ours. In contrast, Absolute Zero and SPIRAL use continuous online training, while Socratic-Zero relies on a fixed external teacher for curriculum generation, making its evolution dynamics different from ours and computationally prohibitive to extend. For a fair comparison, we evaluate all baselines under their standard configurations, and additionally introduce a variant with fewer evolution iterations, R-Diverse*, to match the evolution rounds of R-Zero. See App. D for more details.

*Table 1.* Main results on mathematical and general reasoning benchmarks. Full R-Diverse runs for 5 iterations; R-Diverse* denotes training for 3 iterations (matching R-Zero's standard setup); the other baselines are evaluated under their standard configurations. **Bold**: best; underline: second best.

| Model | Math AVG | Overall AVG | Mathematical Reasoning Benchmarks | | | | | | | General Reasoning | | |
|---|---|---|---|---|---|---|---|---|---|---|---|---|
| | | | AMC | Minerva | MATH | GSM8K | Olympiad | AIME25 | AIME24 | SuperGPQA | MMLU-Pro | BBEH |
| *Qwen3-4B-Base* | | | | | | | | | | | | |
| Base Model | 42.58 | 27.10 | 45.70 | 38.24 | 68.20 | 87.79 | 41.04 | 6.15 | 10.94 | 20.88 | 37.38 | 7.57 |
| Absolute Zero | 46.42 | 33.61 | 52.45 | 41.96 | 76.20 | 89.34 | 42.56 | 10.20 | 12.20 | 27.10 | 52.60 | 8.30 |
| SPIRAL | 47.00 | 34.22 | 57.50 | 42.40 | 76.40 | 91.00 | 38.40 | 10.00 | 13.30 | 27.10 | 53.20 | 9.57 |
| R-Zero | 49.07 | 34.64 | 57.27 | 52.94 | **79.60** | 92.12 | 44.59 | 4.27 | 12.71 | 27.55 | 51.53 | 10.42 |
| R-Diverse* | 50.68 | 36.30 | 59.06 | 56.62 | 79.00 | 92.27 | 45.19 | 6.25 | 16.35 | **28.84** | 54.91 | **10.77** |
| R-Diverse | **52.59** | **36.68** | **60.23** | **59.56** | 78.80 | **92.34** | **46.96** | **11.04** | **19.17** | 28.54 | **55.24** | 10.35 |
| *Qwen3-8B-Base* | | | | | | | | | | | | |
| Base Model | 49.18 | 34.49 | 51.95 | 50.00 | 78.00 | 89.08 | 44.74 | 16.67 | 13.85 | 28.33 | 51.80 | 8.63 |
| Absolute Zero | 52.68 | 38.97 | 57.89 | 57.90 | 76.60 | 92.00 | 47.80 | 18.20 | 18.40 | 31.89 | 60.50 | 10.80 |
| R-Zero | 54.69 | 38.73 | 61.67 | 60.66 | 82.00 | 94.09 | 48.89 | 19.17 | 16.35 | 31.38 | 58.23 | 10.60 |
| Socratic-Zero | 56.10 | 39.15 | 63.70 | 52.40 | 81.20 | 87.30 | **55.10** | **24.60** | **28.40** | 30.10 | 60.90 | 9.50 |
| R-Diverse* | 55.40 | 40.34 | 59.30 | 63.24 | 80.60 | 94.31 | 49.19 | 17.71 | 23.44 | **32.56** | 61.49 | 11.92 |
| R-Diverse | **56.46** | **40.75** | **66.02** | **66.18** | 82.20 | **94.39** | 49.78 | 12.19 | 24.48 | 32.52 | **61.79** | **12.23** |

**Evaluation.** We evaluate on seven mathematical and three general reasoning benchmarks (App. E). We report pass@1 accuracy with greedy decoding for all benchmarks except AMC and AIME, where we use mean@32 following prior work (Huang et al., 2025).

## 4.2. Main Results

**Overall.** Table 1 summarizes the main results on seven mathematical and three general reasoning benchmarks. Across both model scales, R-Diverse achieves the best performance on both Math AVG and Overall AVG.

**Mathematical Reasoning.** R-Diverse delivers the best Math AVG among all prior self-play baselines at both scales. On Qwen3-4B-Base, it boosts Math AVG from 42.58 (Base) to 52.59 (**+10.01**) and surpasses the strongest prior baseline R-Zero by **+3.52**. On Qwen3-8B-Base, it improves Math AVG from 49.18 (Base) to 56.46 (**+7.28**) and exceeds R-Zero by **+1.77**. Moreover, extending evolution from 3 iterations (R-Diverse*) to 5 iterations yields additional gains on both scales (50.68→52.59; 55.40→56.46), supporting sustained improvement rather than early plateauing.

**General Reasoning.** The gains extend beyond math. On Qwen3-4B-Base, R-Diverse raises Overall AVG from 27.10 (Base) to 36.68 (**+9.58**) and outperforms R-Zero by **+2.04**. On Qwen3-8B-Base, it achieves the best Overall AVG (40.75), surpassing R-Zero by **+2.02** and even outperforming Socratic-Zero (**+1.60**), which leverages external API-based question generation.

## 5. Analysis

### 5.1. Ablation Study

💡 *Every R-Diverse component matters.*

*Table 2.* Fine-grained ablation study on R-Diverse components (Qwen3-4B-Base). $\Delta$: difference from full R-Diverse.

| Method | Math AVG | $\Delta$ |
|---|---|---|
| R-Diverse (Full) | **52.59** | — |
| *Ablation on MAP* | | |
| w/o MAP | 49.62 | –2.97 |
| – w/o Max- & Mean-Similarity Penalty | 49.88 | –2.71 |
| – w/o Max-Similarity Penalty | 50.60 | –1.99 |
| – w/o Mean-Similarity Penalty | 50.64 | –1.95 |
| – w/o Memory Replay | 51.18 | –1.41 |
| *Ablation on SAM* | | |
| w/o SAM | 50.50 | –2.09 |
| – w/o Representation Abstraction | 50.79 | –1.80 |
| – w/o Embedding-based Similarity Computation | 51.05 | –1.54 |

To dismantle the identified illusions, R-Diverse relies on both MAP and SAM. We conduct a fine-grained ablation on Qwen3-4B-Base (Table 2).

**Ablation on MAP.** Removing MAP causes the largest drop ($52.59 \rightarrow 49.62$), confirming that global, memory-based repulsion is critical for dispelling the *Local Diversity Illusion*. Within MAP, max- and mean-similarity terms contribute comparably (each $\Delta \approx -2.0$), while dropping both is worse ($\Delta = -2.71$), suggesting they prevent two complementary failures: direct history recycling ($P_{\max}$) and drifting back into dense, previously explored regions ($P_{\mean}$). Finally, replay matters when MAP actively shifts the data distribution: without replay, performance drops by 1.41 points, suggesting increased forgetting.

**Ablation on SAM.** Disabling SAM degrades performance ($\Delta = -2.09$), indicating that combating the *Surface Diversity Illusion* requires going beyond surface overlap. Both abstraction steps are necessary: removing representation abstraction or disabling the embedding-based similarity computation causes substantial drops (1.80 and 1.54 points). This supports our claim that canonicalizing problems into

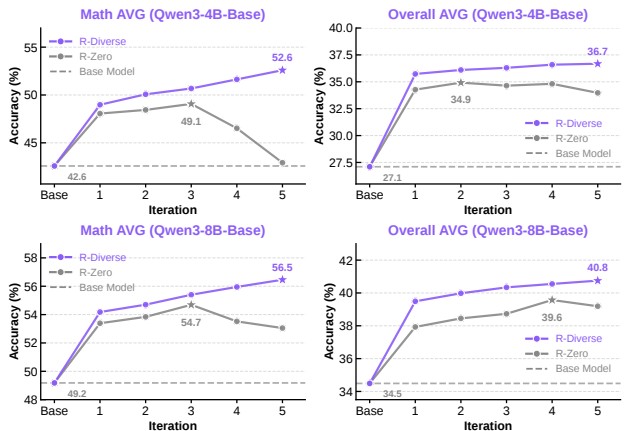

*Figure 3.* Performance across iterations on different metrics and model scales. R-Diverse achieves monotonic improvement across all settings, while R-Zero collapses after iteration 3-4. Stars indicate peak performance.

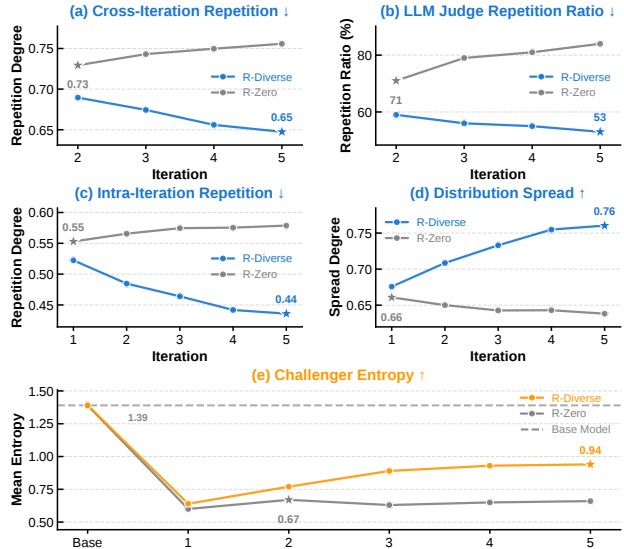

*Figure 4.* Multi-dimensional diversity analysis. **(a-b) Cross-Iteration Repetition**: R-Zero exhibits increasing historical recycling (confirmed by both SAM embedding similarity and the LLM judge), while R-Diverse reduces it. **(c-d) Intra-Iteration Diversity**: R-Diverse maintains lower repetition and higher distribution spread. **(e) Policy Dynamics**: R-Diverse recovers Challenger entropy, avoiding the low-entropy collapse observed in R-Zero. (↑: higher is better; ↓: lower is better.)

solver-level procedures and measuring similarity in a semantic embedding space are jointly required to identify and penalize skill-level repetition.

## 5.2. Sustainability Analysis

### 💡 *R-Diverse achieves sustained improvement.*

A critical question is whether R-Diverse can sustain improvement across multiple evolution iterations. As previewed in Figure 1(d) and detailed in Figure 3, we present the performance trajectory over five iterations across two metrics (Math AVG and Overall AVG) and two model scales (Qwen3-4B-Base and Qwen3-8B-Base).

R-Diverse demonstrates **sustained, monotonic improvement** over five evolution iterations across both model scales and both metrics. On Qwen3-4B, it steadily improves to 52.6 Math AVG and 36.7 Overall AVG, while R-Zero collapses after iteration 3 (49.1→42.9 by iteration 5). The same trend holds on Qwen3-8B: R-Diverse continues to improve to 56.5 Math AVG and 40.8 Overall AVG, whereas R-Zero peaks early and then degrades (down to 53.1 Math AVG / 39.2 Overall AVG at iteration 5). These results suggest that R-Diverse provides a more reliable self-play recipe that scales across model sizes.

We attribute this robustness to the synergy between MAP and SAM: MAP counters the *Local Diversity Illusion* by penalizing historical recycling, while SAM counters the *Surface Diversity Illusion* by identifying skill-level repetition beyond superficial variation. Gains on Overall AVG are smaller than on Math AVG, likely because the self-play curriculum is generated in the mathematical domain.

## 5.3. Diversity Analysis

### 💡 *R-Diverse reduces Local and Surface Diversity Illusions.*

To rigorously verify the effectiveness of R-Diverse in mitigating the Diversity Illusion, we conduct a multi-dimensional analysis focusing on three perspectives: **Cross-Iteration**, **Intra-Iteration**, and **Policy Dynamics**, where five distinct metrics are employed in total to capture the full dynamics of the evolutionary process.

**Metrics.** For **Cross-Iteration**, we quantify historical recycling using (1) *Cross-Iteration Repetition*, computed from the max- and mean-cosine similarity between SAM embeddings of new questions and the memory bank, and (2) *LLM Judge Repetition Ratio*, where GPT-4o judges whether a new question is semantically equivalent to its top-3 nearest historical neighbors. For **Intra-Iteration**, we quantify within-iteration collapse using (3) *Intra-Iteration Repetition* (average pairwise cosine similarity of SAM embeddings) and (4) *Distribution Spread* (average distance to the embedding centroid). For **Policy Dynamics**, we report (5) *Challenger Entropy* over 200 rollouts. Formal definitions are in App. C.

**Stable Diversity.** Figure 4 presents the iteration trajectories of these metrics. Regarding **Cross-Iteration** (Figure 4a, b), R-Zero exhibits increasing historical recycling, with the LLM-judge duplicate ratio rising from 71% to 84%. In contrast, R-Diverse reduces this ratio (59%→53%), consistent

*Table 3.* Cross-iteration evaluation. Each cell shows the pass rate (%) of the Solver (columns) on questions from the Challenger at different iterations (rows). Highlighted cells indicate the pass rate of the target solver used for challenger training (i.e., $\mathcal{S}_{\phi_{k-1}}$ on $\mathcal{D}_{\text{Iter } k}$), which stays around the 50% uncertainty sweet spot.

| Dataset | Base | Solver | | | | |
|---|---|---|---|---|---|---|
| | | Iter 1 | Iter 2 | Iter 3 | Iter 4 | Iter 5 |
| $\mathcal{D}_{\text{Iter } 1}$ | 47.5 | 55.0 | 57.5 | 58.5 | 58.0 | 60.5 |
| $\mathcal{D}_{\text{Iter } 2}$ | 44.0 | 50.5 | 54.0 | 52.5 | 56.5 | 55.0 |
| $\mathcal{D}_{\text{Iter } 3}$ | 46.5 | 47.5 | 49.5 | 51.5 | 53.0 | 56.5 |
| $\mathcal{D}_{\text{Iter } 4}$ | 40.0 | 43.5 | 44.0 | 48.0 | 51.5 | 51.0 |
| $\mathcal{D}_{\text{Iter } 5}$ | 39.0 | 41.0 | 39.5 | 43.0 | 47.0 | 50.5 |

with MAP mitigating the *Local Diversity Illusion*. Regarding **Intra-Iteration** (Figure 4c, d), R-Diverse maintains lower repetition and higher spread, indicating that SAM better captures skill-level diversity beyond superficial variation, mitigating the *Surface Diversity Illusion*. Finally, the **Policy Dynamics** (Figure 4e) show that R-Diverse recovers Challenger entropy (0.64→0.94), while R-Zero remains low-entropy, suggesting that R-Diverse encourages sustained exploration rather than premature exploitation.

## 5.4. Curriculum Learning Preservation

💡 ***R-Diverse preserves curriculum learning.***

A natural concern is whether the diversity-promoting mechanisms in R-Diverse interfere with the uncertainty-based curriculum learning central to self-play. To test this, we construct five evaluation sets $\{\mathcal{D}_{\text{Iter } k}\}_{k=1}^{5}$ by sampling 200 questions from the Challenger at each iteration, and obtain ground-truth labels using GPT-4o.

Table 3 shows two key findings. First, for any fixed Solver (columns), the pass rate generally decreases on later-iteration sets (rows), e.g., Iter 1 drops from 55.0% on $\mathcal{D}_{\text{Iter } 1}$ to 41.0% on $\mathcal{D}_{\text{Iter } 5}$, indicating that the generated curriculum becomes progressively harder. Second, the highlighted diagonal stays close to 50% (i.e., $\mathcal{S}_{\phi_{k-1}}$ on $\mathcal{D}_{\text{Iter } k}$), showing that the Challenger continues to target the uncertainty sweet spot even with MAP and SAM enabled.

Taken together, R-Diverse preserves uncertainty-based difficulty targeting while improving the coverage of training signals, enabling a harder and more diverse curriculum for Solver training. Additional qualitative analyses (question evolution across iterations and examples of surface diversity illusion) are provided in Appendix G.

# 6. Related Work

## 6.1. Self-Play and Self-Evolving LLMs

Self-play, popularized in game AI (e.g., AlphaZero) (Silver et al., 2017; 2018), has recently been adapted to bootstrap reasoning in LLMs without human-curated data (Tao et al., 2024). Beyond verifiable settings such as code generation with execution feedback (Lin et al., 2025; Wang et al., 2025b; Zhao et al., 2025), recent frameworks (e.g., R-Zero (Huang et al., 2025), SPIRAL (Liu et al., 2025)) co-evolve a question generator and a solver using model-based signals (e.g., consistency/self-verification). However, sustained improvement remains challenging: training often becomes unsustainable and may collapse after early gains (Chen et al., 2024; Kuba et al., 2025; Shumailov et al., 2024; Dohmatob et al., 2024). Our work complements these efforts by identifying a key failure mode, **Diversity Illusion**: (i) diversity is enforced only within the current batch, causing cross-iteration mode cycling, and (ii) diversity is measured by surface-form overlap, missing repetition of the underlying reasoning skills. We formalize this as *Local* and *Surface* Diversity Illusions, and propose MAP and SAM to address them.

## 6.2. Memory-Augmented Agents

Memory has become a key ingredient for building LLM agents with longer-term context and continual adaptation (Hu et al., 2025; Fang et al., 2025). Prior work explores diverse memory forms (e.g., logs, vector stores, parametric memory) and uses them to improve *inference* or *single-agent* learning, e.g., for long-horizon interaction and self-reflection (Park et al., 2023; Packer et al., 2023; Shinn et al., 2023; Yao et al., 2023). In contrast, we integrate memory into the **self-play optimization loop**: MAP uses a persistent memory bank as a *repulsive regularizer* to penalize similarity to historical questions, directly targeting the *Local Diversity Illusion*. We further use memory replay to stabilize Solver training under the distribution shift induced by sustained exploration.

## 6.3. Auto-formalization and Semantic Abstraction

Auto-formalization translates natural-language mathematics into rigorous formal systems (e.g., Isabelle/HOL, Lean), reducing linguistic ambiguity and enabling precise checking (Wu et al., 2022; Jiang et al., 2022; Wang et al., 2024a; Xin et al., 2025; Chen et al., 2025b;a). However, theorem proving and formal data collection remain costly, limiting their use inside large-scale self-play loops. An alternative is to use general-purpose code (e.g., Python) as a lightweight semantic proxy (Ni et al., 2023; Chen et al., 2021; Yang et al., 2024), which is often employed for execution-based verification. We repurpose code as a **semantic bottleneck for measurement**: SAM canonicalizes questions into solver-level procedures and computes similarity in a code-embedding space, enabling skill-aware diversity estimation and mitigating the *Surface Diversity Illusion* that evades lexical metrics.

# 7. Conclusion

In this work, we identify Diversity Illusion as a key failure mode of collapse in self-play reasoning and propose R-Diverse to dismantle it. By enforcing diversity through Memory-Augmented Penalty and Skill-Aware Measurement, our framework achieves state-of-the-art performance and sustainable self-improvement. A current limitation is that Skill-Aware Measurement relies on code as a semantic bottleneck; while effective for reasoning-centric tasks, it may not generalize to domains that are difficult to formalize. Future work will explore more universal semantic representations to extend the power of R-Diverse to the broader landscape of general capabilities.

# Acknowledgements

This work was supported by the Prevention and Control of Emerging and Major Infectious Diseases—National Science and Technology Major Project (Grant No. 2025ZD01901600) and by the Beijing Natural Science Foundation (Grant No. L252035). We also thank the anonymous reviewers and the area chair for their constructive feedback during the review process.

# Impact Statement

This paper presents work whose goal is to advance the field of Machine Learning by improving the stability and diversity of self-play training for reasoning-oriented language models. By mitigating failure modes such as diversity illusion, the proposed approach can help researchers develop models that learn more reliably from self-generated data, potentially reducing reliance on large-scale human annotation and enabling more systematic study of self-improvement dynamics. Potential societal impacts are mixed. On the positive side, more robust reasoning models may support applications in education, scientific assistance, and accessibility by enabling more accurate multi-step problem solving. On the negative side, improved reasoning capability could be misused for academic dishonesty or for producing more convincing misleading content, and iterative self-play training may increase computational and energy costs if scaled or deployed irresponsibly. While our work does not introduce fundamentally new classes of capabilities beyond existing large language models, it contributes to making such systems more effective, and we therefore encourage careful evaluation, transparency about limitations, and adherence to relevant safety and usage policies. To support reproducibility and scrutiny, we plan to release our code upon acceptance, together with documentation and evaluation scripts, and we encourage responsible downstream use.

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

# Appendix

# A. Experimental Details

This section provides the key configurations used in R-Diverse. All experiments are conducted on $8\times$ NVIDIA H20 GPUs with BF16 precision and FlashAttention 2 for acceleration.

## A.1. Hyperparameter Settings

**Solver Training**

- Global Batch Size: 128

- Learning Rate: $1 \times 10^{-6}$

- Weight Decay: $1 \times 10^{-2}$

- KL Penalty Coefficient ($\lambda_{KL}$): $1 \times 10^{-2}$

- Max Steps: 15

- Number of Rollouts: 5

- Rollout Temperature: 1.0

- Rollout Top-p: 0.99

**Challenger Training**

- Global Batch Size: 128

- Learning Rate: $1 \times 10^{-6}$

- Weight Decay: $1 \times 10^{-2}$

- KL Penalty Coefficient ($\lambda_{KL}$): $1 \times 10^{-2}$

- Max Steps: 5

- Number of Rollouts: 4

- Rollout Temperature: 1.0

- Rollout Top-p: 0.99

**MAP and SAM Calculation**

- Repetition Penalty Weight ($\alpha$): 1.0

- Memory-Augmented Penalty Weight ($\beta$): 1.0

- Max-Mean Mixing Coefficient ($\gamma$): 0.5

- Max-Similarity Threshold ($\tau_{\mathrm{max}}$): 0.5

- Mean-Similarity Threshold ($\tau_{\mathrm{mean}}$): 0.25

- Memory Replay Ratio ($\rho$): 0.3

- Code Generation Model: Qwen2.5-Coder-7B (Temperature: 0)

- Code Embedding Model: Jina-Code-Embeddings-1.5B

## A.2. Prompt Templates

This section presents all prompts used in R-Diverse training and analysis. All prompts remain identical to R-Zero (Huang et al., 2025), except for the Code Generation Prompt (designed for SAM) and the LLM Judge Repetition Ratio Prompt (for measuring cross-iteration repetition).

---

**Solver Prompt Template**

**System Prompt:**
Please reason step by step, and put your final answer within \boxed{}.
**User Prompt:**
{problem statement}

---

**Challenger Prompt Template**

**System Prompt:**
You are an expert competition-math problem setter. FIRST, in your private scratch-pad, think step-by-step to design a brand-new, non-trivial problem. The problem could come from any field of mathematics, including but not limited to algebra, geometry, number theory, combinatorics, prealgebra, probability, statistics, and calculus. Aim for a difficulty such that fewer than 30% of advanced high-school students could solve it. Avoid re-using textbook clichés or famous contest problems. THEN, without revealing any of your private thoughts, output exactly the following two blocks:
```
<question>
{The full problem statement on one or more lines}
</question>
\boxed{final answer}
```
Do NOT output anything else—no explanations, no extra markup.
**User Prompt:**
Generate one new, challenging reasoning question now. Remember to format the output exactly as instructed.

---

**Code Generation Prompt Template (for SAM)**

This prompt instructs the model to extract the abstract mathematical logic from a question and encode it as a pure Python function.

- - - - - - - - - - - - - - - - - - - - - - - - - - - - - - - - - - - - - - - - - - - - -

# Role
You are a Strict Python Logic Encoder for Mathematical Problems.
# Objective
Convert the user's Math Question into a **pure, comment-free Python solver function**. The code must represent the abstract solution logic used to solve or prove the problem.
# Critical Constraints (MUST FOLLOW)

1. **Strictly NO Comments**: The output code must NOT contain any comments (`#` or `"""`). No docstrings. Pure code only.

2. **Function Signature (Separating Data from Logic)**: You must define a single function named `solver`. **Extract all numerical values** from the question and put them as **default arguments** in the function definition (e.g., `def solver(n1=35, n2=94):`). The function body should ONLY contain variable definitions (unknowns) and operations/assertions.

3. **Variable Naming (Abstract Only)**:
   **FORBIDDEN**: Semantic names (e.g., `apple`, `speed`, `price`).
   **REQUIRED**: Arguments: `n1`, `n2`, `n3`...; Variables: `x`, `y`, `z`, `a`, `b`, `c`; Constants/Limits: `k`, `m`.

4. **Handling Proof/Proving Questions**: If the question asks to "Prove X", the code must construct the hypothesis using `sympy` symbols and return a **Boolean** value (`True` if the proof holds). Do not print "Proven". Return the result of the logical check (e.g., `return sympy.simplify(lhs - rhs) == 0`).

---

## Code Generation Prompt Template (for SAM) — Continued

5. **Robustness for Flawed Input**: If the input question is ambiguous, logically flawed, or poorly phrased, **do not complain**. Infer the most likely intended mathematical logic and generate code for that valid interpretation.

6. **Output Format**: Output ONLY the code block enclosed in `` tags.

**# Few-Shot Examples**
**User Input (Arithmetic):** "John has 5 apples and buys 3 more. How many does he have?"
**Output:**

```

def solver(n1=5, n2=3):
    res = n1 + n2
    return res

```

**User Input (Algebra):** "A farm has chickens and rabbits. There are 35 heads and 94 legs. Find the number of rabbits."
**Output:**

```

from sympy import symbols, Eq, solve
def solver(n1=35, n2=94):
    x, y = symbols('x y')
    eq1 = Eq(x + y, n1)
    eq2 = Eq(2*x + 4*y, n2)
    sol = solve((eq1, eq2), (x, y))
    return sol[y]

```

**User Input (Proof/Theorem):** "Prove that $(a + b)^2$ is always equal to $a^2 + 2ab + b^2$."
**Output:**

```

from sympy import symbols, simplify
def solver():
    a, b = symbols('a b')
    expr1 = (a + b)**2
    expr2 = a**2 + 2*a*b + b**2
    return simplify(expr1 - expr2) == 0

```

**User Input (Flawed/Noisy Question):** "Calculate velocity if distance is 100, voltage is 20V... um... and time is 20."
**Output:**

```

def solver(n1=100, n2=20):
    v = n1 / n2
    return v

```

**# Begin Task**
Input Question: {question}

## LLM-as-Judge Prompt Template

This prompt is used to recheck the answer. We use GPT-4o as the judge model.

- - - - - - - - - - - - - - - - - - - - - - - - - - - - - - - - - - - - - - - - - - - - - -

**System Prompt:**
You are a math answer checker.
**User Prompt:**
Hi, there is an answer: {answer}, and the ground truth answer is: {response}, please check whether the answer is correct or not, and return the **only** Yes or No.

---

**LLM Judge Repetition Ratio Prompt Template**

This prompt instructs GPT-4o to determine whether a new problem is semantically equivalent to its top-k nearest historical neighbors.

- - - - - - - - - - - - - - - - - - - - - - - - - - - - - - - - - - - - - - - - - - - -

You are a strict mathematics problem similarity detector. Your task is to determine if a NEW problem is essentially a duplicate of any REFERENCE problem.
**## Strict Duplicate Criteria** (if ANY is true, answer DUPLICATE):

1. **Same mathematical technique**: Both require the same core method (e.g., both use quadratic formula, both use integration by parts, both use modular arithmetic).

2. **Same problem structure**: The problems have the same logical structure with different numbers/names.

3. **Same concept testing**: Both test the same mathematical concept (e.g., both test understanding of geometric series).

4. **Trivial transformation**: One can be obtained from another by simple substitution of variables, numbers, or context.

5. **Isomorphic problems**: Different story/context but mathematically identical (e.g., "chickens and rabbits" vs "cars and motorcycles" counting legs/wheels).

**## REFERENCE PROBLEMS** (from previous iterations):
`{reference_problems}`
**## NEW PROBLEM** (from current iteration):
`{new_problem}`
**## Important**:

- Be STRICT: When in doubt, lean towards DUPLICATE.

- Focus on the MATHEMATICAL ESSENCE, not surface differences.

- Two problems about "different topics" (geometry vs algebra) can still be DUPLICATES if they use the same technique.

Answer with ONLY one word - "NOVEL" or "DUPLICATE":

## B. Computational Efficiency

This section presents the computational overhead introduced by SAM. Despite the additional code generation and embedding computation, R-Diverse does not introduce significant overhead compared to R-Zero. In fact, replacing BLEU-based clustering (a CPU-intensive $O(n^2)$ process) with code embedding similarity slightly reduces computation time. We further optimize the original R-Zero Challenger training pipeline through time-multiplexing. These optimizations enable R-Diverse to complete one evolution iteration in approximately 6 hours on Qwen3-4B-Base, reducing training time by 20% compared to R-Zero (7.5 hours).

## C. Metric Definitions

This section provides the formal mathematical definitions for the diversity metrics used in Section 5.3.

**Cross-Iteration Repetition.** This metric measures how much a new set of questions overlaps with historically generated questions stored in the memory bank $\mathcal{M}$. For a set of questions $\mathcal{Q}_t$ generated at iteration $t$, we compute:

$$\text{Cross-Iter-Rep}(\mathcal{Q}_t, \mathcal{M}) = \frac{1}{|\mathcal{Q}_t|} \sum_{q \in \mathcal{Q}_t} \left( \frac{1}{2} \max_{e \in \mathcal{M}} \cos(\phi(q), e) + \frac{1}{2} \cdot \frac{1}{|\mathcal{M}|} \sum_{e \in \mathcal{M}} \cos(\phi(q), e) \right). \tag{12}$$

This combines both max-cosine similarity (detecting direct repetition) and mean-cosine similarity (detecting distributional overlap).

**Intra-Iteration Repetition.** This metric measures the internal homogeneity within a single iteration of generated questions.

For a set $\mathcal{Q}_t$, we compute the average pairwise cosine similarity:

$$\text{Intra-Iter-Rep}(\mathcal{Q}_t) = \frac{1}{|\mathcal{Q}_t|} \sum_{q_i \in \mathcal{Q}_t} \frac{1}{|\mathcal{Q}_t| - 1} \sum_{q_j \in \mathcal{Q}_t, j \neq i} \cos(\phi(q_i), \phi(q_j)). \tag{13}$$

**LLM Judge Repetition Ratio.** To provide another view of cross-iteration repetition, we employ GPT-4o as an independent judge. For each question $q$ in the current iteration, we retrieve its top-3 nearest neighbors from the memory bank based on cosine similarity. The judge determines whether $q$ is semantically equivalent to any of these neighbors. The repetition ratio is the fraction of questions judged as duplicates:

$$\text{LLM-Rep-Ratio}(\mathcal{Q}_t) = \frac{1}{|\mathcal{Q}_t|} \sum_{q \in \mathcal{Q}_t} \mathbb{I}[\text{Judge}(q, \text{Top3}(q, \mathcal{M})) = \text{DUPLICATE}]. \tag{14}$$

The prompt template used for the LLM judge is provided in the "LLM Judge Repetition Ratio Prompt Template" shown above.

**Distribution Spread.** To provide another view of intra-iteration repetition, this metric quantifies how dispersed the generated questions are in the semantic embedding space. We compute the centroid of all embeddings and measure the average distance from each question to this centroid:

$$\text{Spread-Degree}(\mathcal{Q}_t) = \frac{1}{|\mathcal{Q}_t|} \sum_{q \in \mathcal{Q}_t} \left\| \phi(q) - \bar{\phi} \right\|_2, \quad \text{where } \bar{\phi} = \frac{1}{|\mathcal{Q}_t|} \sum_{q \in \mathcal{Q}_t} \phi(q). \tag{15}$$

A higher spread indicates greater diversity in the generated question distribution.

**Challenger Entropy.** This metric probes the exploratory nature of the Challenger's generation policy by measuring the token-level entropy averaged over multiple rollouts. For $N$ rollouts, each producing a sequence of tokens $\{t_1, t_2, \ldots, t_L\}$:

$$\text{Entropy} = \frac{1}{N} \sum_{n=1}^{N} \frac{1}{L_n} \sum_{l=1}^{L_n} H(p(\cdot|t_{<l})), \tag{16}$$

where $H(p(\cdot|t_{<l})) = -\sum_{v \in \mathcal{V}} p(v|t_{<l}) \log p(v|t_{<l})$ is the entropy of the next-token distribution conditioned on the preceding context $t_{<l}$, and $\mathcal{V}$ denotes the vocabulary. Higher entropy indicates a more exploratory policy.

## D. Baseline Details

We compare R-Diverse against the following representative methods:

- **Base Model**: The pre-trained Qwen3-4B-Base or Qwen3-8B-Base model without any post-training, serving as the lower-bound reference.

- **R-Zero** (Huang et al., 2025): The foundational self-play framework for reasoning LLMs that we build upon. It introduces uncertainty-driven curriculum learning with BLEU-based repetition penalty. R-Zero serves as our primary baseline.

- **Absolute Zero** (Zhao et al., 2025): A self-play framework that leverages code execution for verification. The model generates both problems and solutions, using execution feedback to validate correctness. This represents a strong tool-augmented baseline.

- **Socratic-Zero** (Wang et al., 2025a): A self-play variant that leverages external APIs for curriculum question generation. This represents methods that trade evolution autonomy for quality.

- **SPIRAL** (Liu et al., 2025): A self-play approach that employs a zero-sum game formulation to bootstrap the training between the Challenger and the Solver.

# E. Evaluation Benchmarks

## E.1. Mathematical Reasoning Benchmarks

- **AMC**: A collection of problems from American Mathematics Competitions (AMC 10/12), representing standard high school competition mathematics.

- **MATH** (Hendrycks et al., 2021): A comprehensive dataset of 12,500 challenging competition mathematics problems spanning seven subjects (Prealgebra, Algebra, Number Theory, Counting and Probability, Geometry, Intermediate Algebra, and Precalculus).

- **GSM8K** (Cobbe et al., 2021): A benchmark of 8,500 grade school math word problems requiring multi-step arithmetic reasoning, serving as a test of basic mathematical reasoning capabilities.

- **Olympiad-Bench** (He et al., 2024): A challenging benchmark comprising highly difficult problems from Chinese and International Mathematical Olympiads, aimed at testing the limits of LLM reasoning capabilities.

- **Minerva** (Lewkowycz et al., 2022): A benchmark evaluating the model's ability to handle formal scientific notation and solve complex STEM-related questions requiring integration of mathematical and scientific reasoning.

- **AIME24 & AIME25**: Problems from the 2024 and 2025 American Invitational Mathematics Examinations, representing the most recent and likely uncontaminated competition problems.

## E.2. General Reasoning Benchmarks

- **MMLU-Pro** (Wang et al., 2024b): An upgraded MMLU variant featuring more challenging questions, expanded answer choices (10 vs. 4), and heightened reasoning demands to better distinguish advanced models.

- **SuperGPQA** (Du et al., 2025): An evolution of the GPQA dataset featuring difficult graduate-level questions across scientific domains, containing 285 different disciplines which are unsearchable. This benchmark is designed to minimize data contamination and retrieval shortcuts.

- **BBEH** (Kazemi et al., 2025): A selected set of difficult Big-Bench tasks targeting known LLM weaknesses, such as symbolic reasoning, logical inference, and algorithmic state tracking.

# F. Additional Results

This section provides additional results that complement our main experiments. We first report detailed iteration-wise performance across all five evolution iterations on both Qwen3-4B-Base and Qwen3-8B-Base, and then examine the generalization of R-Diverse to another base model family. We further present a series of ablations that probe the two core components: for SAM, its sensitivity to the choice of code generator and embedding model, and a general-purpose, code-free variant; for MAP, the effect of the memory bank size and alternative finer-grained penalty designs.

## F.1. Qwen3-4B-Base Results

*Table 4.* Iteration-wise performance of Qwen3-4B-Base on mathematical reasoning benchmarks. **Bold**: best result across iterations.

| Iteration | Math AVG | AMC | Minerva | MATH | GSM8K | Olympiad | AIME25 | AIME24 |
|---|---|---|---|---|---|---|---|---|
| Base Model | 42.58 | 45.70 | 38.24 | 68.20 | 87.79 | 41.04 | 6.15 | 10.94 |
| Iter 1 | 48.98 | 51.80 | 57.72 | 76.20 | **92.72** | 43.26 | 11.67 | 9.48 |
| Iter 2 | 50.07 | 56.95 | **59.93** | 77.40 | 91.96 | 43.11 | 2.19 | 18.96 |
| Iter 3 | 50.68 | 59.06 | 56.62 | **79.00** | 92.27 | 45.19 | 6.25 | 16.35 |
| Iter 4 | 51.63 | 57.03 | 56.99 | 77.40 | 91.21 | 45.33 | **16.04** | 17.40 |
| Iter 5 | **52.59** | **60.23** | 59.56 | 78.80 | 92.34 | **46.96** | 11.04 | **19.17** |

**Key Observations:**

- The first iteration yields the largest single-step improvement (+6.40 on Math AVG), suggesting that the initial self-play phase efficiently addresses low-hanging fruit in reasoning capability.

- AMC and Minerva show particularly strong gains (+14.53 and +21.32 respectively from base to iteration 5), indicating that R-Diverse effectively generates challenging competition-level problems.

## F.2. Qwen3-8B-Base Results

*Table 5.* Iteration-wise performance of Qwen3-8B-Base on mathematical reasoning benchmarks. **Bold**: best result across iterations.

| Iteration | Math AVG | AMC | Minerva | MATH | GSM8K | Olympiad | AIME25 | AIME24 |
|---|---|---|---|---|---|---|---|---|
| Base Model | 49.18 | 51.95 | 50.00 | 78.00 | 89.08 | 44.74 | 16.67 | 13.85 |
| Iter 1 | 54.18 | 55.39 | 65.07 | 79.60 | 93.86 | 49.19 | 13.96 | 22.19 |
| Iter 2 | 54.70 | 60.62 | 63.24 | 82.60 | 93.56 | 49.33 | 14.37 | 19.17 |
| Iter 3 | 55.40 | 59.30 | 63.24 | 80.60 | 94.31 | 49.19 | **17.71** | 23.44 |
| Iter 4 | 55.95 | 59.45 | 65.07 | **83.00** | 93.71 | **50.22** | 13.75 | **26.46** |
| Iter 5 | **56.46** | **66.02** | **66.18** | 82.20 | **94.39** | 49.78 | 12.19 | 24.48 |

**Key Observations:**

- R-Diverse achieves consistent monotonic improvement across all five iterations, with Math AVG steadily increasing from 49.18 to 56.46, demonstrating stable self-evolution without performance collapse.

- AMC and Minerva show particularly strong gains (+14.07 and +16.18 respectively from base to iteration 5), indicating that R-Diverse effectively generates challenging competition-level problems.

## F.3. Generalization to Other Base Model Families

To verify that the gains of R-Diverse are not specific to the Qwen3 family, we follow R-Zero (Huang et al., 2025) and evaluate on the OctoThinker series (Wang et al., 2025c), a non-Qwen3 model family also adopted by R-Zero, at both the 3B and 8B scales. Table 6 reports the Math AVG trajectory.

*Table 6.* Cross-family generalization on the OctoThinker series. We report Math AVG over five iterations for R-Zero and R-Diverse on OctoThinker-3B and OctoThinker-8B. **Bold**: peak result across iterations for each method.

| Iteration | OctoThinker-3B | | OctoThinker-8B | |
|---|---|---|---|---|
| | R-Zero | R-Diverse | R-Zero | R-Diverse |
| Base Model | 26.64 | 26.64 | 36.41 | 36.41 |
| Iter 1 | 27.76 | 28.34 | 37.80 | 38.24 |
| Iter 2 | 28.20 | 28.97 | 38.23 | 38.67 |
| Iter 3 | **29.32** | 30.02 | **38.52** | 39.24 |
| Iter 4 | 28.57 | **30.90** | 38.03 | 39.74 |
| Iter 5 | 27.11 | 30.82 | 37.54 | **40.07** |

**Key Observations:**

- R-Diverse consistently outperforms R-Zero at every iteration on both scales, improving the peak Math AVG by **+1.58** (30.90 vs. 29.32) on OctoThinker-3B and **+1.55** (40.07 vs. 38.52) on OctoThinker-8B.

- Consistent with the trend on Qwen3, R-Zero peaks early (iteration 3) and then degrades on both scales, whereas R-Diverse achieves more stable evolution. The smaller 3B model peaks at iteration 4 while the 8B model sustains improvement through iteration 5, echoing R-Zero's observation that self-play capacity scales with base-model size.

- These results confirm that the diversity-driven improvements of R-Diverse generalize beyond the Qwen3 family, indicating that mitigating the Diversity Illusion is a model-agnostic benefit rather than an artifact of a specific base model.

## F.4. SAM Component Sensitivity

A potential concern is that the gains of SAM may largely reflect the quality of the specific code generator and embedding model used, rather than the skill-aware measurement itself. To examine this, we vary both components of SAM, replacing the default code generator (Qwen2.5-Coder-7B) with a smaller (Qwen2.5-Coder-3B) and a different-family (DeepSeek-Coder-6.7B) coder, and replacing the default embedder (Jina-Code-Embeddings-1.5B) with four alternatives of different sizes and families. Table 7 reports the resulting Math AVG on Qwen3-4B-Base.

*Table 7.* Sensitivity of SAM to the choice of code generator and embedding model on Qwen3-4B-Base. All variants are compared against the default SAM ($\Delta$ default) and the no-SAM baseline ($\Delta$ w/o SAM = 50.50).

| Code Generator | Embedding Model | Math AVG | $\Delta$ default | $\Delta$ w/o SAM |
|---|---|---|---|---|
| Qwen2.5-Coder-7B | Jina-Code-1.5B (default) | **52.59** | — | +2.09 |
| *Varying the code generator* | | | | |
| Qwen2.5-Coder-3B | Jina-Code-1.5B | 52.08 | –0.51 | +1.58 |
| DeepSeek-Coder-6.7B | Jina-Code-1.5B | 51.74 | –0.85 | +1.24 |
| *Varying the embedding model* | | | | |
| Qwen2.5-Coder-7B | Jina-Code-0.5B | 51.83 | –0.76 | +1.33 |
| Qwen2.5-Coder-7B | CodeT5+-110M | 52.12 | –0.47 | +1.62 |
| Qwen2.5-Coder-7B | Qwen3-Embed-0.6B | 51.94 | –0.65 | +1.44 |

**Key Observations:**

- Every tested configuration outperforms the no-SAM baseline (50.50), with the smallest gain still at **+1.24**. This indicates that the benefit of SAM is robust to the choice of both the code generator and the embedding model.

- Using smaller models has only a minor impact on performance (a drop of at most $-0.85$ from the default), making SAM well-suited for deployment under compute constraints.

## F.5. A General-Purpose Variant of SAM

Our default SAM instantiation maps each question to canonical Python code, which is naturally suited to the math-centric self-play setting but, as noted in our Conclusion, may be less applicable to domains that are difficult to formalize as code. We emphasize, however, that the central idea of SAM is *not* tied to code: it is to abstract away surface variation *before* measuring similarity. To demonstrate this, we construct a general-purpose, code-free variant of SAM. Concretely, we replace the code generator (Qwen2.5-Coder-7B) with a general instruction model (Qwen2.5-7B-Instruct) that paraphrases each question into a canonical, structured statement, and replace the code embedder (Jina-Code-Embeddings-1.5B) with a general-purpose text encoder (Qwen3-Embed-0.6B). All other components are unchanged.

*Table 8.* A general-purpose, code-free variant of SAM on Qwen3-4B-Base. Replacing the code generator and code embedder with a general instruction model and a general-purpose text encoder still improves over w/o SAM, indicating that SAM's gain stems from its skill-aware abstraction principle rather than from code specifically. $\Delta$: difference from w/o SAM.

| Configuration | Abstraction Model | Embedding Model | Math AVG | $\Delta$ |
|---|---|---|---|---|
| w/o SAM | — | — | 50.50 | — |
| General-Purpose SAM | Qwen2.5-7B-Instruct | Qwen3-Embed-0.6B | 51.54 | +1.04 |
| Code-Based SAM (default) | Qwen2.5-Coder-7B | Jina-Code-Embeddings-1.5B | **52.59** | **+2.09** |

**Key Observations:**

- Even without any code-specific model, the general-purpose variant improves Math AVG from 50.50 (w/o SAM) to **51.54 (+1.04)**. This shows that the benefit of SAM stems from its skill-aware abstraction principle rather than from code generation per se.

- The default code-based SAM remains the strongest (52.59, +2.09), indicating that for math-centric self-play, canonicalizing questions into the code is a particularly effective realization of the abstraction principle.

- Together, these results show that the principle of SAM can transfer to a code-free instantiation, offering a practical path toward extending SAM to domains that are hard to formalize. We leave this extension as future work, consistent with the limitation discussed in our Conclusion.

## F.6. Memory Bank Size

MAP maintains a persistent memory bank that grows across iterations. A natural question is how much history is needed and whether storing the entire history is necessary. We compare the full memory against bounded windows that retain only the most recent iterations, as well as the no-MAP baseline. Table 9 reports Math AVG on Qwen3-4B-Base.

*Table 9.* Effect of the memory bank size in MAP on Qwen3-4B-Base. We compare the full memory against bounded windows of recent iterations and the no-MAP baseline. Δ Full: difference from the full-memory default; Δ no MAP: difference from the no-MAP baseline.

| Setting | Math AVG | Δ Full | Δ no MAP |
|---|---|---|---|
| Full memory (default) | **52.59** | — | +2.97 |
| Last 4 iterations | 52.37 | −0.22 | +2.75 |
| Last 3 iterations | 51.89 | −0.70 | +2.27 |
| No MAP | 49.62 | −2.97 | — |

**Key Observations:**

- The bulk of the gain comes from recent history: retaining only the last 3 iterations still improves over no-MAP by +2.27. This shows that MAP is effective even under a bounded memory budget.

- Performance degrades as the window shrinks further, confirming that cross-iteration memory is the key driver of MAP's benefit.

## F.7. Finer-Grained Penalty Designs

The mean-similarity penalty $P_{\text{mean}}$ in MAP (Eq. 8, repeated below for convenience) treats the memory bank as a single distribution and penalizes the average similarity of a new question to *all* historical samples:

$$P_{\text{mean}}(q, \mathcal{M}) = \frac{1}{|\mathcal{M}|} \sum_{e \in \mathcal{M}} \cos(\phi(q), e). \tag{17}$$

While this global-mean formulation is simple and stable, it ignores the local geometry of the explored region: a question may be highly redundant with one dense cluster of past questions yet still receive a low penalty if the rest of the memory bank is far away. We therefore further explore two finer-grained alternatives that account for the structure of $\mathcal{M}$.

**Density-Aware Penalty (KNN).** Instead of averaging over the entire memory bank, we penalize the average similarity to only the $K$ nearest historical neighbors of $q$, making the penalty sensitive to local density:

$$P_{\text{KNN}}(q, \mathcal{M}) = \frac{1}{K} \sum_{e \in \mathcal{N}_K(q)} \cos(\phi(q), e), \tag{18}$$

where $\mathcal{N}_K(q)$ denotes the set of $K$ memory entries with the highest cosine similarity to $\phi(q)$. This sharply penalizes questions that fall into a densely explored neighborhood, even when the global mean similarity is low.

**Cluster-Aware Penalty (K-Means).** Alternatively, we partition the memory bank into $C$ clusters $\{\mathcal{M}_1, \dots, \mathcal{M}_C\}$ via K-Means and penalize the similarity to cluster centroids $\{\mu_1, \dots, \mu_C\}$, weighted by cluster size to emphasize over-represented regions:

$$P_{\text{KMeans}}(q, \mathcal{M}) = \sum_{c=1}^{C} \frac{|\mathcal{M}_c|}{|\mathcal{M}|} \cos(\phi(q), \mu_c). \tag{19}$$

This provides a coarser, distribution-level view that retains the stability of the global mean while better capturing the multiple modes of the memory distribution.

**Key Observations:**

*Table 10.* Exploration of finer-grained penalty designs on Qwen3-4B-Base. Using the global-mean penalty of R-Diverse as the reference, we evaluate a Density-Aware (KNN) and a Cluster-Aware (K-Means) variant. The KNN-based formulation attains the highest peak but is more sensitive to its hyperparameter, while the K-Means-based formulation is stable across configurations with more modest gains. $\Delta$: difference from the global-mean reference.

| Method | Math AVG | $\Delta$ |
|---|---|---|
| *Reference* | | |
| Global Mean | 52.59 | — |
| *Density-Aware (KNN)* | | |
| $K = 50$ | 51.97 | −0.62 |
| $K = 100$ | **53.38** | **+0.79** |
| $K = 200$ | 52.73 | +0.14 |
| *Cluster-Aware (K-Means)* | | |
| $C = 10$ | 52.67 | +0.08 |
| $C = 20$ | 52.73 | +0.14 |
| $C = 40$ | 52.85 | +0.26 |

- The Density-Aware (KNN) penalty achieves the strongest result, lifting Math AVG from 52.59 to **53.38 (+0.79)** at $K = 100$. This confirms that accounting for local density, rather than only the global mean, can more precisely identify and repel redundant exploration.

- The KNN variant is, however, sensitive to $K$: an overly small neighborhood ($K = 50$) even underperforms the reference ($−0.62$), as it overfits to a few nearest points and becomes noisy.

- The Cluster-Aware (K-Means) penalty yields consistent but more modest gains ($+0.08$ to $+0.26$) and is robust across the number of clusters $C$, offering a stable middle ground between the global mean and the more aggressive KNN formulation.

- Overall, these results indicate that the global-mean penalty adopted in R-Diverse is an effective default that is simple to implement and hyperparameter-free, while more sophisticated finer-grained penalty designs offer a promising direction for further unlocking the potential of R-Diverse.

## G. Case Analysis

### G.1. Question Evolution Across Iterations

To illustrate how the R-Diverse framework generates progressively more challenging questions across training iterations, we present three representative case studies. These examples demonstrate the natural evolution of question complexity as the Challenger model learns to create problems that challenge an increasingly capable Solver.

#### G.1.1. CASE STUDY 1: NUMBER THEORY PROBLEMS

Table 11 shows the evolution of number-theoretic problems related to divisors and prime factorization. The questions progressively incorporate more constraints and require deeper mathematical reasoning.

**Analysis:** Iteration 1 presents a straightforward divisibility condition. By Iteration 2, multiple constraints are introduced (digit products, perfect cubes, prime divisibility restrictions). Iteration 3 requires understanding of periodic decimal expansions combined with primality constraints on individual digits. Iteration 4 introduces the concept of multiplicative order and combines it with perfect square and modular conditions. Finally, Iteration 5 creates a novel mathematical concept ("super-divisible") requiring analysis of the totient function's behavior across all divisors.

#### G.1.2. CASE STUDY 2: SEQUENCE PROBLEMS

Table 12 demonstrates the evolution of sequence-based problems, showing how the framework generates increasingly sophisticated definitions and constraints.

**Analysis:** Iteration 1 uses a simple recursive formula with a direct computation task. Iteration 2 introduces a greedy construction based on coprimality constraints. Iteration 3 defines a novel sequence property ("harmonic") and asks about counting such sequences. Iteration 4 adds a sophisticated GCD condition that creates a partially ordered structure. Iteration

*Table 11.* Evolution of Number Theory Questions Across Iterations

| Iter. | Question |
|---|---|
| 1 | What is the smallest positive integer $n$ such that the number of divisors of $n^2$ is twice the number of divisors of $n$? |
| 2 | Find the smallest positive integer $n$ such that the product of its digits is a perfect square, and $n$ itself is a perfect cube. Additionally, $n$ must not be divisible by any prime less than 10. |
| 3 | Find the smallest positive integer $n$ such that the decimal representation of $\frac{1}{n}$ is a repeating decimal with a period of exactly 6 digits, where each digit in the repeating cycle is a prime number. |
| 4 | Let $S$ be the set of all prime numbers. For each prime $p \in S$, define $f(p)$ as the smallest positive integer $k$ such that $2^k \equiv 1 \pmod{p}$. Determine the smallest prime $p$ for which $f(p)$ is a perfect square and $f(p) \equiv 1 \pmod 4$. |
| 5 | A positive integer $n$ is called *super-divisible* if for every divisor $d$ of $n$, the number $\phi(n) \bmod d$ is a prime number, where $\phi(n)$ denotes Euler's totient function. Find the smallest positive integer $n$ that is both super-divisible and has the property that $\phi(n)$ is divisible by the number of its divisors. |

5 combines branching sequence construction rules with a counting problem over constrained sequence spaces.

### G.1.3. CASE STUDY 3: COMBINATORIAL COUNTING PROBLEMS

Table 13 illustrates the evolution of combinatorial counting problems, showing how constraints and problem structures become increasingly sophisticated.

**Analysis:** Iteration 1 presents a basic combination counting problem. Iteration 2 adds an implicit constraint requiring reverse engineering of the population size. Iteration 3 introduces set partition with multiple sum constraints that require careful divisibility analysis. Iteration 4 combines letter arrangement with ordering constraints, non-adjacency conditions, and conditional probability. Iteration 5 defines a sophisticated counting measure (inversions) over a constrained permutation subspace.

These case studies demonstrate that R-Diverse naturally generates questions with increasing complexity along multiple dimensions: (1) the number of constraints, (2) the depth of mathematical concepts required, (3) the novelty of problem definitions, and (4) the sophistication of the solution approach needed.

### G.2. Surface Diversity Illusion Examples

The Surface Diversity Illusion refers to the phenomenon where questions appear textually diverse (as measured by low BLEU scores) but are semantically equivalent or highly similar (as measured by high embedding similarity scores). This section provides concrete examples from our generated question bank.

### G.2.1. EXAMPLE 1: IDENTICAL PROBLEMS WITH DIFFERENT PHRASING

The following questions, generated independently across different iterations, ask exactly the same mathematical problem but with varied phrasing:

**Analysis:** These three questions have relatively low pairwise BLEU scores (due to different sentence structures: "Find the number..." vs. "How many ways can..." vs. "How many ways are there..."), yet they are mathematically identical. Traditional diversity metrics based on n-gram overlap would classify these as diverse questions, while our code-based semantic similarity correctly identifies them as duplicates.

### G.2.2. EXAMPLE 2: STRUCTURAL TEMPLATES WITH PARAMETER VARIATIONS

The following questions share identical mathematical structure but differ only in numerical parameters:

**Analysis:** While the numerical values differ, all three questions require applying the Chinese Remainder Theorem to find

*Table 12.* Evolution of Sequence Problems Across Iterations

| Iter. | Question |
|---|---|
| 1 | A sequence of positive integers $a_1, a_2, a_3, \ldots$ satisfies $a_1 = 1$ and for all $n \geq 1$, $a_{n+1} = a_n^2 + a_n$. Find the number of positive divisors of $a_{2023}$. |
| 2 | A sequence of positive integers $a_1, a_2, \ldots$ is defined as follows: $a_1 = 1$, and for each $n \geq 1$, $a_{n+1}$ is the smallest integer greater than $a_n$ that is relatively prime to all previous terms in the sequence. Determine the value of $a_{100}$. |
| 3 | A sequence of positive integers $a_1, a_2, \ldots, a_n$ is called *harmonic* if for every positive integer $k$ with $1 \leq k \leq n$, the number $k \cdot a_k$ is a perfect square. Let $H(n)$ be the number of harmonic sequences of length $n$. Find the smallest positive integer $n$ such that $H(n)$ is a multiple of 100. |
| 4 | Consider a sequence of positive integers $a_1, a_2, \ldots, a_n$ such that for any $i \neq j$, the greatest common divisor $\gcd(a_i, a_j)$ is either 1 or $a_i$. Let $S$ be the set of all possible values of the sum $a_1 + a_2 + \cdots + a_n$. Find the smallest positive integer $n$ such that the number of elements in $S$ is greater than 100. |
| 5 | A sequence of positive integers $a_1, a_2, \ldots, a_{10}$ is defined such that each term $a_n$ is either the smallest positive integer not already in the sequence or the smallest positive integer not already in the sequence that shares a common divisor greater than 1 with the previous term $a_{n-1}$. Determine the number of distinct sequences that can be formed if $a_{10} \leq 10$. |

consecutive integers satisfying simultaneous modular conditions. The underlying solver code would be nearly identical, differing only in constant values. This represents a common mode of "diversity" that provides no genuine training signal variety.

### G.2.3. EXAMPLE 3: NARRATIVE WRAPPERS AROUND IDENTICAL MATHEMATICS

Questions can be wrapped in different narrative contexts while encoding the same mathematical problem:

**Analysis:** Both questions are graph coloring problems on path graphs. Question A counts 3-colorings of a 10-vertex path, while Question B counts 10-colorings of a 20-vertex path. The narrative elements ("small town," "peculiar town," "mayor," "painting competition") create surface diversity, but the mathematical core—counting proper vertex colorings of a path—remains essentially the same.

### G.2.4. IMPLICATIONS FOR TRAINING DATA QUALITY

These examples illustrate why SAM is crucial for measuring true diversity:

- **BLEU-based filtering fails**: Questions A, B, and C in Table 14 might pass BLEU-based deduplication due to different phrasings.

- **Template-based generation creates illusory diversity**: Varying only numerical parameters (Table 15) inflates question counts without adding mathematical variety.

- **Narrative wrapping obscures redundancy**: Story-based problems (Table 16) may appear creative but often encode standard textbook problems.

SAM directly addresses these issues by comparing the underlying skills of the generated questions rather than the surface-level text.

*Table 13.* Evolution of Combinatorial Counting Problems Across Iterations

| Iter. | Question |
|---|---|
| 1 | How many distinct ways are there to choose 3 members from a committee of 10, where the order of selection does not matter? |
| 2 | In a small town, there are $n$ residents who decide to form clubs such that each club contains exactly 4 members. If the number of ways to form these clubs is 126 times the number of ways to form a single club, find the smallest possible value of $n$. |
| 3 | Find the number of ways to partition the set $\{1, 2, \ldots, 2023\}$ into three non-empty subsets $A$, $B$, and $C$ such that the sum of the elements in $A$ is twice the sum of the elements in $B$, and the sum of the elements in $C$ is three times the sum of the elements in $B$. |
| 4 | How many distinct arrangements can be made from the letters of the word "MATHEMATICS", if we require that the vowels (A, E, I) always appear together in an arrangement, but each vowel must be in alphabetical order, and no two identical letters can be adjacent? Additionally, calculate the probability that the last letter is an M given that the vowel group is included. |
| 5 | Let $P$ be the set of all permutations of the numbers $1, 2, \ldots, 10$. For a permutation $\sigma \in P$, define $f(\sigma)$ as the number of inversions in $\sigma$. Find the sum of $f(\sigma)$ over all $\sigma \in P$ such that the first 5 elements of $\sigma$ are in increasing order. |

*Table 14.* Semantically Identical Questions with Textual Variations

| ID | Question |
|---|---|
| A | Find the number of ways to arrange the letters in the word "MATHEMATICS" such that no two vowels are adjacent. |
| B | How many ways can you arrange the letters in the word "MATHEMATICS" such that no two vowels are adjacent? |
| C | How many ways are there to arrange the letters in "MATHEMATICS" such that no two vowels are adjacent? |

*Table 15.* Structurally Identical Questions with Different Parameters

| ID | Question |
|---|---|
| A | Find the smallest positive integer $n$ such that $n$ is divisible by 3, $n + 1$ is divisible by 4, and $n + 2$ is divisible by 5. |
| B | Find the smallest positive integer $n$ such that $n$ is divisible by 5, $n + 1$ is divisible by 7, and $n + 2$ is divisible by 11. |
| C | Find the smallest positive integer $n$ such that $n$ is divisible by 3, $n + 1$ is divisible by 5, and $n + 2$ is divisible by 7. |

*Table 16.* Different Narratives Encoding Similar Mathematics

| ID | Question |
|---|---|
| A | In a small town, there are 10 houses in a row. Each house can either be painted red, blue, or green. However, no two adjacent houses can be painted the same color. How many different ways can the 10 houses be painted? |
| B | In a peculiar town, there are 20 houses numbered from 1 to 20. The mayor assigns each house a unique color from a palette of 10 colors, ensuring no two houses with consecutive numbers share the same color. How many different ways can the mayor assign these colors? |

