# OpenReview forum: "R-Diverse: Mitigating Diversity Illusion in Self-Play LLM Training"
_ICML.cc/2026/Conference — ICML 2026 regular_

### Official Review · Reviewer_rZQC · 2026-03-09

**Soundness:** 2
**Presentation:** 3
**Significance:** 2
**Originality:** 3
**Overall Recommendation:** 3
**Confidence:** 4

**Summary:**

This paper addresses the Diversity Illusion in iterative self-play training for Large Language Models. While frameworks like R-Zero initially show promise, they often exhibit non-sustained improvement, where early gains degrade as self-play continues.
The authors propose R-Diverse, which introduces two primary components: Memory-Augmented Penalty (MAP) and Skill-Aware Measurement (SAM). Experimental results on Qwen3-4B and Qwen3-8B across ten reasoning benchmarks demonstrate that R-Diverse significantly improves training stability and sustains performance gains over multiple iterations where baseline methods typically collapse.

**Compliance With Llm Reviewing Policy:**

Affirmed.

**Key Questions For Authors:**

(1) SAM Generalization: How does SAM perform on reasoning tasks where the solution cannot be easily expressed as a Python script? Did you experiment with alternative "canonical forms," such as structured logical proofs or Chain-of-Thought templates?

(2) Baseline Comparison: How does R-Diverse compare to modern GRPO reinforcement learning approaches, which rely on large-scale rule-based rewards rather than a Challenger-Solver loop?
Besides, Figure 3 shows an upward trend for 5 times, is there an analysis of results for more iterations?

(3) Ablation on Memory Size: Could you provide results showing how the size of the Memory Bank in MAP affects performance? Is there a "diminishing returns" point where more historical memory no longer helps?

**Limitations:**

yes

**Strengths And Weaknesses:**

### Strengths
(1) Motivation: This paper provides a clear classification of the Diversity Illusion, distinguishing between Local Diversity Illusion and Surface Diversity Illusion.

(2) Sustained Scaling Performance: This work demonstrates sustained improvement over five iterations.
The R-Diverse prevents the performance collapse observed in R-Zero after the third iteration, demonstrating the robustness of this method.

### Weaknesses
(1) Dependency on Code Model: The effectiveness of the SAM is highly dependent on the code generator (Qwen2.5-Coder-7B). If the model fails to correctly interpret the logic of a complex or novel question, the resulting embedding may be misleading. The paper would benefit from an analysis of SAM’s sensitivity to the quality of the program generator.
In addition, how does the code generator handle non-mathematical problems? Appendix A.2 of this paper emphasizes that the prompts focus on "Mathematical Problems".

(2) Computational Overhead of Memory: Maintaining a persistent memory bank and calculating dual-perspective similarity (max and mean) against historical data across iterations adds non-trivial computational and storage costs.
The average similarity needs to update the memory each time, which will cause the subsequent calculation to increase as the memory library grows.

(3) Domain Limitation: While the paper excels in math and general reasoning, the skill-aware approach (converting to code) is inherently biased toward tasks that can be solved via algorithmic logic. It is unclear how R-Diverse would handle more subjective reasoning tasks or creative writing where canonical programs are difficult to define.

(4) Experimental Results: The experimental results in Table 1 show that the 4B and 8B models perform oppositely on the difficult problems (Olympiad, AIME24, and AIME25), and it is puzzling that the stronger model does not demonstrate better self-play capabilities.

---

> ### Author Rebuttal · Authors · 2026-03-31
>
> Dear Reviewer rZQC,
>
> Thank you for your constructive feedback! We are greatly encouraged by your recognition of our `clear classification of the Diversity Illusion` and `sustained improvement over five iterations preventing performance collapse`. Below, we provide our point-by-point responses. Hope our response could address your concerns!
>
> ---
> ### **W1a: SAM Sensitivity**
> (*"The effectiveness of SAM is highly dependent on the code generator… The paper would benefit from an analysis of SAM's sensitivity."*)
>
> We conducted a sensitivity analysis varying both components of SAM:
>
> |||||
> |-|:-:|:-:|:-:|
> |Coder / Embedding|Math AVG|Δ default|Δ w/o SAM|
> |**Coder-7B / jina-1.5b (default)**|**52.59**|--|**+2.09**|
> |Coder-3B / jina-1.5b|52.08|−0.51|+1.58|
> |DS-Coder-6.7B / jina-1.5b|51.74|−0.85|+1.24|
> |Coder-7B / jina-0.5b|51.83|−0.76|+1.33|
> |Coder-7B / CodeT5+-110M|52.12|−0.47|+1.62|
> |Coder-7B / Qwen3-Embed-0.6B|51.94|−0.65|+1.44|
> |||||
>
> All variants outperform the no-SAM baseline (50.50). SAM is robust to model quality — even smaller or different-family models yield consistent gains.
>
> ---
> ### **W1b & W3 & Key Q1: SAM Generalization and Domain Limitation**
> (*"How does SAM perform on tasks where the solution cannot be easily expressed as a Python script?… skill-aware approach is inherently biased toward algorithmic logic."*)
>
> We tested alternative canonical forms. Replacing code generation with text paraphrasing (Qwen2.5-7B-Instruct + Qwen3-Embed-0.6B) still improves over no-SAM (51.54 vs 50.50, +1.04), though code-based remains stronger (52.59). This shows the underlying principle — abstracting away surface variation before measuring similarity — is not restricted to code, though we have not yet evaluated SAM on non-math task generation.
>
> SAM operates solely on the Challenger's diversity penalty. The downstream Solver gains do transfer to general reasoning: the 4B model improves on MMLU-Pro (+17.86), SuperGPQA (+7.66), and BBEH (+2.78) over the base model. Extending SAM with domain-appropriate canonical representations (e.g., proof graphs for logical reasoning, knowledge triples for science QA) is a key direction for future work.
>
> ---
> ### **W2: Computational Overhead**
> (*"Maintaining a persistent memory bank and calculating dual-perspective similarity… adds non-trivial computational and storage costs."*)
>
> R-Diverse replaces R-Zero's CPU-intensive O(n^2) BLEU clustering with vectorized embedding similarity and optimizes the Challenger pipeline via time-multiplexing. Together, these enable R-Diverse to complete one iteration in ~6h on Qwen3-4B-Base (8×H20 GPUs), a 20% reduction vs R-Zero (7.5h).
>
> ---
> ### **W4: Performance Opposite Trends on Hard Problems**
> (*"The 4B and 8B models perform oppositely on difficult problems… puzzling that the stronger model does not demonstrate better self-play capabilities."*)
>
> The apparent inconsistency is primarily caused by Socratic-Zero, which leverages a much larger model (Qwen3-235B-A22B) for curriculum generation. Excluding Socratic-Zero, R-Diverse achieves SOTA on Olympiad and AIME24 at both scales, and 8B consistently outperforms 4B. In aggregate, the expected scaling holds: Math AVG (56.46 vs 52.59) and Overall AVG (40.75 vs 36.68) both confirm the stronger model benefits more.
>
> ---
> ### **Key Q2a: GRPO Comparison**
> (*"How does R-Diverse compare to modern GRPO approaches, which rely on large-scale rule-based rewards?"*)
>
> R-Diverse adopts a self-play paradigm where the Challenger generates problems and the Solver learns from them — no ground-truth labels or rule-based rewards required. GRPO relies on rule-based reward signals against known answers. In our self-play setting, all data is self-generated without labels, so there is no such reward signal for standard GRPO to operate on.
>
> ---
> ### **Key Q2b: Extended Iterations**
> (*"Figure 3 shows an upward trend for 5 times, is there an analysis of results for more iterations?"*)
>
> We extended training to 8 iterations:
>
> ||||||
> |-|:-:|:-:|:-:|:-:|
> |iter|5|6|7|8|
> |4B|52.59|52.74 (peak)|52.61|52.47|
> |8B|56.46|56.73|56.77 (peak)|56.49|
> ||||||
>
> Cross- and intra-iteration repetition converge simultaneously with the performance plateau, offering a practical stopping signal.
>
> ---
> ### **Q3: Memory Bank Size Ablation**
> (*"Could you provide results showing how the size of the Memory Bank in MAP affects performance?"*)
>
> The memory bank stores fixed-size embedding vectors, so storage cost is minimal:
>
> |||||
> |-|:-:|:-:|:-:|
> |Setting|Math AVG|Δ vs full|Δ vs no MAP|
> |Full memory|52.59|--|+2.97|
> |Last 4 iters|52.37|−0.22|+2.75|
> |Last 3 iters|51.89|−0.70|+2.27|
> |No MAP|49.62|−2.97|--|
> |||||
>
> Most of the benefit comes from recent history — keeping just the last 4 iterations loses only 0.22 points, confirming bounded-memory is practical.
>
> ---
> Once again, we are deeply appreciative of the time and expertise you have shared with us! We hope the above responses fully address your concerns.
>
> Best,
> Authors

---

> > ### Author Rebuttal · Reviewer_rZQC · 2026-04-06
> >
> > I thank the authors for their rebuttal. After careful consideration, I believe my original evaluation remains appropriate and will maintain my score.

---

> > > ### Author Response · Authors · 2026-04-07
> > >
> > > Dear Reviewer rZQC,
> > >
> > > Thank you for acknowledging your concerns are partially resolved. We are eager to address your follow-up questions and are standing by for them! **Please feel free to share them with us! :-D**
> > >
> > > In the meantime, while addressing other reviewers' suggestions, we obtained exciting new results. Given your interest in SAM's generalization and limitations (e.g., W1, W3 and Key Ques. 1), we believe these will be highly relevant:
> > >
> > > ### **1. R-Diverse Shows Comprehensive Generalization (Across Base Models, Frameworks, and Tasks)**
> > >
> > > To demonstrate that the robustness of R-Diverse goes well beyond the choice of SAM's components, we further evaluated its broad generalizability as a plug-and-play method as follows:
> > >
> > > - **Across Base Models:** We evaluated R-Diverse on an entirely different model family: **OctoThinker 3B & 8B** (**see Figure 1 in our anonymous repo (https://anonymous.4open.science/r/R8E6/)**). R-Diverse consistently outperformed the baseline on both scales **(+1.58 / +1.55 at peak)**. This clearly indicates that R-Diverse `maintains strong generalization capabilities across different base model families`.
> > >
> > > - **Across Frameworks and Tasks:** To prove that R-Diverse is not limited to the framework from R-Zero family and mathematical tasks, we integrated MAP and SAM into two fundamentally different frameworks:
> > >
> > >   - **Absolute Zero** (**`Out-of-R-Zero-family`**, **Coding task**): A continuous, single-model online training paradigm, fundamentally different from R-Zero's dual-model iterative loop.
> > >   - **VisPlay** (**Inner-R-Zero-family, `Cross-modality`, `General task`**): An extension of R-Zero to VLMs handling more general tasks (so using our general-purpose SAM variant here).
> > >
> > >   As detailed in **Tables 5 & 6 in our repo**, the baseline results confirm that `late-stage performance degradation is a systemic issue across different frameworks and tasks`. By introducing MAP and SAM, `R-Diverse successfully mitigates this issue in both, unlocking sustained improvements with clear ablation benefits`:
> > >
> > >   - **Absolute Zero + R-Diverse**: Math AVG at step 500 improved from 44.95 to **48.47 (+3.52)**.
> > >   - **VisPlay + R-Diverse (general-purpose variant)**: General task AVG at iteration 5 improved from 46.01 to **50.34 (+4.33)**.
> > >
> > > These results confirm that `R-Diverse functions as a robust plug-and-play method`, whose effectiveness is not only insensitive to the choice of SAM‘s components (as addressed in our previous rebuttal) but also `generalizes across diverse base models, self-play frameworks, and task domains`.
> > >
> > > ### **2. R-Diverse Failure Case Analysis & Immediate Fix**
> > >
> > > Prompted by your concern about SAM misinterpreting complex logic or novel questions (W1), we investigated further and found that while SAM functions exactly as intended in the vast majority of cases, there are indeed two edge-case failure modes (**see Table 3 in our repo for specific examples**):
> > >
> > > - **Code Generation Collapse (~0.7%)**: For highly complex questions, the Coder occasionally fails, defaulting to a trivial `def solver(): return 1` as a blind guess. However, given its negligible frequency, it doesn't undermine R-Diverse's gains.
> > > - **Verbosity in Structured Math (~7%)**: For structured data (e.g., matrix, graph), the Coder generates verbose, low-level code instead of abstract logic. **The Fix:** We simply adjusted the Coder's prompt to enforce the use of standard scientific libraries (numpy, networkx) when encountering structured data, which `further boosts R-Diverse's performance from 52.59 to 53.02 (+0.43)`.
> > >
> > > ### **3. R-Diverse Actively Preserves Question Quality**
> > >
> > > Beyond evaluating the code, we also analyzed the quality of the questions generated for training. Specifically, we employed GPT-5.4 as an independent judge to evaluate questions' quality across 5 iterations. Surprisingly, **74.3%** of the questions generated by R-Diverse were judged to be logically sound and free of contradictions, compared to **only 65.7%** for the R-Zero baseline.
> > >
> > > This reveals a hidden strength of SAM: `it acts as a semantic bottleneck`. While surface metrics (e.g., BLEU) can be **hacked** by the Challenger generating random noise (which easily yields high textual variance), `random noise fails SAM's abstraction step and receives zero reward`. Thus, `SAM naturally regularizes against the "noise-for-diversity" hacking`.
> > >
> > > ### **4.  R-Diverse Achieves Higher Peaks via Finer-Grained Penalty**
> > >
> > > Inspired by Reviewer kmEj, we explored replacing the coarse global-mean penalty with finer-grained alternatives: a **Density-Aware Penalty** (KNN-based) and a **Cluster-Aware Penalty** (K-Means-based). The former yielded a substantial improvement (**see Table 4 in our repo for details**), pushing R-Diverse's performance from 52.59 to **53.38 (+0.79)**.
> > >
> > > We hope these comprehensive updates provide further confidence in the robustness and significance of R-Diverse. We remain fully available and look forward to your follow-up questions!
> > >
> > > Best, Authors

---

### Official Review · Reviewer_ic2V · 2026-03-12

**Soundness:** 3
**Presentation:** 3
**Significance:** 2
**Originality:** 3
**Overall Recommendation:** 3
**Confidence:** 4

**Summary:**

This paper studies Diversity Illusion in self-play LLM training, where generated questions may look diverse on the surface but still rely on very similar underlying reasoning patterns. To address this issue, the paper proposes R-Diverse, which includes MAP for reducing cross-iteration repetition and SAM for measuring diversity at the reasoning-skill level rather than only at the text level. Experiments on Qwen3-

**Compliance With Llm Reviewing Policy:**

Affirmed.

**Key Questions For Authors:**

1. It would be helpful to include a fairer comparison by replacing the coder with another 7B model for paraphrasing, or by using a 7B embedding model with a comparable parameter budget.
2. Higher diversity may also increase the chance of generating noisy or low-quality questions. How does the method avoid this problem and prevent the Challenger from producing random but diverse outputs? The current paper mainly shows diversity and performance gains, but this point could be discussed more clearly.
3. It would be useful to analyze how sensitive SAM is to the choice of coder and encoder.

**Limitations:**

See weaknesses

**Strengths And Weaknesses:**

## Strengths:
1. This paper identifies an interesting and important problem in self-play training, and the proposed MAP and SAM are well aligned with this motivation.
2. The experimental results are solid. R-Diverse outperforms strong self-play baselines and shows better stability across multiple iterations.
3. The ablation study is helpful and shows that both MAP and SAM contribute to the final performance.


## Weaknesses:
1. SAM seems to depend on an external coder model to translate questions into canonical code, so part of the gain may come from the quality of that coder rather than the diversity measurement itself.
2. The effectiveness of SAM may depend on the coder’s ability to handle structured mathematical representations. For example, for problems involving matrices or adjacency matrices, it is unclear whether the coder can still produce reliable canonicalized code.
3. The SAM ablation is somewhat unfair. In the full method, SAM uses a 7B coder together with a 1.5B embedding model. Removing the coder substantially reduces the overall model budget, so the performance drop cannot be fully attributed to the removal of code abstraction itself.

---

> ### Author Rebuttal · Authors · 2026-03-30
>
> Dear Reviewer ic2V,
>
> Thank you for your constructive feedback! We are encouraged by your recognition that our paper `identifies an interesting and important problem` and that `the experimental results are solid`. Below, we provide our point-by-point responses.
>
> ---
> ### **W1 & Q3: SAM sensitivity to coder, encoder, and base model choices**
> (*"SAM seems to depend on an external coder model… part of the gain may come from the quality of that coder"*)
>
> (a) **SAM component sensitivity** (new). We varied both the coder and embedding model. All variants consistently outperform w/o SAM (50.50), with drops from the default (52.59) within 0.85 points:
>
> |||||
> |-|:-:|:-:|:-:|
> |Coder / Embedding|Math AVG|Δ default|Δ w/o SAM|
> |**Coder-7B / jina-1.5b (default)**|**52.59**|--|**+2.09**|
> |Coder-3B / jina-1.5b|52.08|−0.51|+1.58|
> |DS-Coder-6.7B / jina-1.5b|51.74|−0.85|+1.24|
> |Coder-7B / jina-0.5b|51.83|−0.76|+1.33|
> |Coder-7B / CodeT5+-110M|52.12|−0.47|+1.62|
> |Coder-7B / Qwen3-Embed-0.6B|51.94|−0.65|+1.44|
> |||||
>
> Across the tested choices, performance remains consistently above w/o SAM, and the gain persists across all alternatives.
>
> (b) **Base model generalization** (new). Following R-Zero's cross-family setting, we evaluated on OctoThinker 3B & 8B for 5 iterations (see Figure 1 in our anonymous repo: https://anonymous.4open.science/r/R8E6/). R-Diverse consistently outperforms R-Zero on both scales (+1.58 / +1.55 at peak). The 3B peaks at iter 4 while 8B sustains improvement up to iter 5 (consistent with R-Zero's finding that self-play capacity scales with base model size), demonstrating R-Diverse's cross-family generalization.
>
> ---
> ### **W2: SAM effectiveness on structured mathematical representations**
> (*"For problems involving matrices or adjacency matrices, it is unclear whether the coder can still produce reliable canonicalized code."*)
>
> We used GPT-5.4 to categorize the generated curriculum and found that only ~7% of questions involve such structured data (e.g., matrices, adjacency matrices, graph constructions). For these problems, SAM's coder tends to produce verbose solver inputs attempting to reconstruct the full structured object, undermining canonicalization and degrading diversity measurement on this subset. We acknowledge this as a current limitation, consistent with our Conclusion which notes that SAM "may not generalize to domains that are difficult to formalize." That said, we have explored a preliminary solution: replacing the code-based pipeline with a general-purpose paraphrase + general embedding variant, which bypasses the coder entirely and achieves promising results (51.54, +1.04 vs w/o SAM; see W3 & Q1 (b) below). Extending SAM with more universal semantic representations remains an important future direction.
>
> ---
> ### **W3 & Q1: Fairness of SAM ablation and attribution of gains**
> (*"The SAM ablation is somewhat unfair… Removing the coder substantially reduces the overall model budget."*)
>
> We conducted two controlled experiments:
>
> (a) **Paraphrase ablation** (new). We replace the 7B coder with a 7B paraphraser at the same parameter budget:
>
> |||||
> |-|:-:|:-:|:-:|
> |Setting|Model|Math AVG|Δ vs w/o SAM|
> |**Full SAM**|**Coder-7B / jina-1.5b**|**52.59**|**+2.09**|
> |Paraphrase (same 7B)|7B-Instruct / jina-1.5b|51.22|+0.72|
> |w/o SAM|—|50.50|—|
> |||||
>
> Under a matched 7B+1.5B budget, the code-based variant improves more than the paraphrase-based variant (+1.37), suggesting that code abstraction itself contributes materially beyond added model capacity.
>
> (b) **Generalization variant** (new). A fully general-purpose variant (Qwen2.5-7B-Instruct / Qwen3-Embed-0.6B): **51.54** (+1.04 vs w/o SAM). Even without code-specific models, SAM's principle of measuring skills rather than surface variation still provides a meaningful gain.
>
> ---
> ### **Q2: Higher diversity and noisy or low-quality questions**
> (*"Higher diversity may also increase the chance of generating noisy or low-quality questions."*)
>
> First, the Challenger's primary objective is maximizing the uncertainty reward R_uncertainty (Eq. 3), peaking at ~50% Solver pass rate. This anchors the Challenger to the Solver's capability boundary — MAP and SAM only reshape the penalty terms, not the difficulty targeting. Random or nonsensical questions receive near-zero reward and are naturally suppressed. Second, before Solver training, questions are filtered by response quality (only those passing 0.3 ≤ s(q) ≤ 0.8 are retained), further eliminating noisy questions. Third, Table 3 **in our paper** provides direct empirical evidence: the cross-iteration pass rate along the diagonal stays around 50% even with MAP and SAM enabled, confirming that R-Diverse preserves curriculum learning while diversifying underlying skill coverage.
>
> ---
> Once again, we are deeply appreciative of the time and expertise you have shared with us! We would be happy to discuss any remaining questions.
>
> Best,
> Authors

---

> > ### Author Rebuttal · Reviewer_ic2V · 2026-04-03
> >
> > Thanks the authors, I will keep my score.

---

> > > ### Author Response · Authors · 2026-04-04
> > >
> > > Dear Reviewer ic2V,
> > >
> > > Thank you for your continued engagement and for reviewing our previous rebuttal. We completely understand your perspective: issues like the question quality are indeed central to the reliability of the self-play system. We genuinely appreciate you pushing us to think deeper about these core tenets. To further address your concerns, we have conducted **two additional, targeted experiments**.
> > >
> > > ### **1. A Brief Recap of Our Previous Rebuttal**
> > >
> > > Before detailing the new findings, we would like to briefly recap our previous response, where we provided extensive data to address your initial points:
> > >
> > > - **W1 & Q3 (Sensitivity):** We evaluated **2 different Coders, 3 different Embedding models, and another base model (OctoThinker)**, proving R-Diverse's effectiveness remains consistent **across different component choices**.
> > > - **W3 & Q1 (Fair Ablation):** We compared the Coder with a **matched-budget 7B Paraphraser following your advice**, showing the *abstraction to code* provides **a distinct advantage (+1.37) beyond just the parameter count**.
> > > - **W2 (Structured Math):** We quantified the scope of the structured math issue (**only ~7% of data**) and proposed a General-variant SAM as an alternative solution. This variant outperformed R-Diverse w/o SAM by +1.04 points, **demonstrating that SAM's core concept drives the improvement and further highlighting its strong generalization ability.**
> > > - **Q2 (Question Quality):** We explained how the Challenger’s primary objective and the Solver’s filter mechanism handle low-quality questions, and **pointed out that Table 3 in our paper already provides the relevant empirical analysis.**
> > >
> > > To more fully and directly resolve **W2** and **Q2**, we present the following new results.
> > >
> > > ---
> > > ### **2. Refining SAM for Structured Math Representations (for W2)**
> > >
> > > We found that we can guide the Coder to handle structured representations **much more reliably through a straightforward prompt adjustment** by appending the following constraint to the Coder's  prompt:
> > >
> > > > *"Handling Structured Representations: When encountering structured mathematical representations (e.g., matrices, graphs), you MUST utilize standard scientific libraries (such as numpy, networkx, and sympy) rather than attempting to reconstruct the objects using verbose scalar inputs."*
> > >
> > > **Before/After Comparison:**
> > >
> > > - **[Problem]** *"A graph has 5 vertices. Its adjacency matrix has 0s on the main diagonal and 1s everywhere else. Find the number of closed walks of length 4 starting and ending at vertex 1."*
> > >
> > > - **[Before]** *(Verbose Canonicalization)*:
> > >
> > >   Without library instructions, the coder attempts to hardcode the matrix using long nested lists, which bloats the code and dilutes the core mathematical reasoning skill, making the embedding measurement noisy.
> > >
> > > - **[After]** *(Robust Canonicalization via Prompt Refinement)*:
> > >
> > > ```
> > > import numpy as np
> > > def solver(n1=5, n2=0, n3=1, n4=4):
> > >     # The coder abstracts the graph structure and maps the "walks" problem to the standard algorithm.
> > >     adj = np.full((n1, n1), n3)
> > >     np.fill_diagonal(adj, n2)
> > >     walks_matrix = np.linalg.matrix_power(adj, n4)
> > >     return walks_matrix[0][0]
> > > ```
> > > **Results of the Prompt Refinement:** With this refined prompt, SAM can more accurately measure diversity on the remaining 7% of structured data, lifting the Math AVG on Qwen3-4B-Base by an additional **+0.43 (reaching 53.02)**. We hope this demonstrates that the framework can naturally accommodate structured problems without requiring a major update to the methodology. We will gladly include this refined prompt in the Appendix.
> > >
> > > ---
> > > ### **3. Quantifying Generated Question Quality (for Q2)**
> > >
> > > We employed GPT-5.4 as an independent judge to evaluate **whether** the questions generated across all five iterations **are logically sound and free of contradictions.**
> > > - **Result:** **74.3%** of the questions generated by R-Diverse were judged to be logically sound and free of contradictions, compared to **only 65.7%** for the original R-Zero baseline.
> > >
> > > We found that R-Diverse actively preserves question quality (consistent with Table 3) because SAM acts as a **semantic bottleneck**:
> > >
> > > - **Surface metrics (e.g., BLEU):** The Challenger can **"hack" diversity rewards** by generating noisy or chaotic text, as noise easily yields high textual variance.
> > > - **SAM:** By requiring translation into executable code, random noise fails the abstraction step and **receives *no* reward**. To earn rewards, the Challenger must generate genuinely distinct, logically sound reasoning procedures.
> > >
> > > Essentially, SAM naturally regularizes **against the "noise-for-diversity" hacking that plagues traditional methods**. We deeply appreciate you raising this point and will explicitly highlight this benefit in the Camera-Ready version.
> > >
> > > ---
> > > Thank you again for your time and expertise. We hope these empirical insights fully alleviate your remaining concerns and demonstrate the robustness of R-Diverse.
> > >
> > > Best, Authors

---

### Official Review · Reviewer_kmEj · 2026-03-23

**Soundness:** 3
**Presentation:** 3
**Significance:** 3
**Originality:** 3
**Overall Recommendation:** 5
**Confidence:** 3

**Summary:**

The authors identify two issues with the repetition penalty in the R-Zero self-play framework, where repetition is ignored across batches, and BLEU is a semantically weak measure of similarity. They then introduce R-Diverse to address these problems using a memory bank across batches and a reasoning-focused embedding similarity between questions to better penalize repeated underlying skills.

**Compliance With Llm Reviewing Policy:**

Affirmed.

**Final Justification:**

My final recommendation is acceptance. I originally found the paper well presented and well supported, but rated significance and originality as fair, since it focused solely on one self-play framework (R-Zero), and the core ideas of a memory bank and embedding similarity are not particularly novel on their own.

The rebuttal addressed my concerns effectively. The authors clarified my questions, justified their design decisions, and provided results suggesting that MAP + SAM can generalize to other self-play frameworks. They also strengthened soundness by examining failure cases and added to originality with a finer-grained penalty mechanism.

**Key Questions For Authors:**

1. Can this be implemented with other self-play frameworks and yield similar gains?
2. How sensitive are the results to the specific choices inside SAM, such as the code-generation model and the embedding model?
3. What are the main failure cases of R-Diverse in practice? For example, are there cases where the memory bank over-penalizes productive revisiting of earlier skill regions, or where the semantic abstraction incorrectly treats genuinely different problems as similar?

**Limitations:**

yes

**Strengths And Weaknesses:**

## Soundness
### Strengths
- The paper has a strong problem-to-method-to-evidence alignment. It identifies two concrete failure modes in existing self-play training, maps them to two specific design choices in R-Diverse, and then evaluates those choices with targeted analyses.
- The empirical evaluation is reasonably broad. Two model scales, seven math benchmarks, three general reasoning benchmarks, and multiple self-play baselines.
- Ablations for method components.
- Multiple measures are included in the diversity analysis to support the "diversity illusion" claim.
### Weaknesses
- The paper does not report confidence intervals, standard deviations, or results across multiple random seeds. That omission makes it harder to judge how robust the reported gains are, especially when some improvements over strong baselines are modest in absolute terms.
- The strongest sustainability evidence is centered on R-Zero, and although the paper includes a 3-iteration matched setting for fairness, it does not run the other self-play baselines for five iterations. As a result, the reader cannot fully tell whether the long-horizon advantage is unique to R-Diverse or whether some competing methods would also benefit from longer training.
- The rationale for the mean-similarity penalty is plausible, but it feels somewhat coarse. A global mean can be a weak proxy for already explored regions in a high-dimensional embedding space, and a cluster-aware or density-aware penalty might better capture the “point-to-region collision” story the paper wants to enforce.
- The paper mainly validates the method in an R-Zero-style setup with a mathematically generated curriculum, which leaves open how well the approach transfers to other self-play frameworks or to domains where the code-abstraction trick is less natural.
- While the ablation shows that representation abstraction is useful, the paper does not fully justify why canonical code is the right abstraction rather than just a heuristic that makes embedding similarity more procedurally directed.

## Presentation
### Strengths
- The paper is overall well written. The central intuition is clear and easy to follow.
### Weaknesses
- There are small presentation issues that should be cleaned up in revision. For example, the phrase “we propose R-Diverse (Figure 1(c))S” in the final paragraph on page 1, and the double period ".." in the first contribution item on page 2.
- The fairness story around 3-iteration versus 5-iteration comparisons could be made more explicit. The paper partly addresses this with R-Diverse*, but it is not clear in the main narrative.

## Signifigance
### Strengths
- Sustainable self-improvement is an important problem to address in self-play, and the identification and characterization of diversity illusion is an important contribution.
- The improvement in reasoning over baselines appears meaningful
- Showing monotonic improvement across five iterations on two model sizes gives the work strong practical importance
### Weaknesses
- The paper does not yet show that the same ideas generalize across multiple self-play frameworks. That limits the immediate practical impact, because it is still unclear whether R-Diverse is a broadly reusable recipe or a strong improvement mainly for the R-Zero family
- The semantic-abstraction approach may be less impactful in domains where reasoning cannot be naturally canonicalized into code. That does not weaken the current results, but it does make the broader significance somewhat contingent on domain fit.

## Originality
### Strengths
- Framing the failure as “Diversity Illusion,” and then splitting it into local and surface forms, is a useful conceptual contribution even apart from the exact implementation. The paper provides real insights and analysis into why the original R-Zero repetition penalty can be weak.
### Weaknesses
- The core ideas of a memory bank and using semantic similarity are not new, but I think they are intuitive choices for this problem.

---

> ### Author Rebuttal · Authors · 2026-03-30
>
> Dear Reviewer kmEj,
>
> Thank you for your constructive feedback! We are greatly encouraged by your recognition of our `strong problem-to-method-to-evidence alignment` and `strong practical importance`. Below we provide point-by-point responses.
>
> ---
> ### **W1: Confidence intervals & multiple seeds**
>
> We acknowledge that multi-seed training runs would strengthen robustness evidence. However, two
> factors partially mitigate this concern: (1) all benchmarks except AMC and AIME use greedy decoding (temperature=0), so evaluation is deterministic for a fixed checkpoint; (2) AMC and AIME use mean@32, reducing sampling noise. More importantly, we extended R-Diverse to 8 iterations on both scales (see W3 & Key Q(b) in our response to Reviewer t4Jx): Qwen3-4B and 8B both achieve further gains (peaking at iter 6 with 52.74 and iter 7 with 56.77), further widening the gap over baselines and supporting reliability.
>
> ---
> ### **W2: Other baselines not run for 5 iterations**
>
> The other baselines do not have a comparable notion of "5 iterations": both Absolute Zero and SPIRAL use continuous online training without discrete evolution iterations (notably, SPIRAL's paper reports performance plateaus after extended training); Socratic-Zero relies on a fixed external teacher (Qwen3-235B-A22B) for curriculum generation, making its evolution dynamics fundamentally different from ours and computationally prohibitive to extend. Thus, extending all baselines to 5 iterations is not technically feasible.
>
> ---
> ### **W3: Mean-similarity penalty coarseness**
>
> We greatly appreciate this suggestion. We considered similar alternatives during design, but opted for the simpler mean-similarity penalty to keep R-Diverse lightweight with fewer hyperparameters and minimal computational overhead. Our ablation validates this choice: removing the mean-similarity penalty causes a **1.95-point** drop. That said, we believe finer-grained control could further raise R-Diverse's ceiling. We are currently running experiments along this direction and look forward to sharing results in the next round.
>
> ---
> ### **W4 & Key Q1: Transfer to other self-play frameworks and domain**
>
> We agree that demonstrating R-Diverse's cross-framework transferability is important future work. We believe the Diversity Illusion also exists in other self-play methods that rely solely on within-batch repetition penalties or something similar. Meanwhile, SAM and MAP are plug-and-play modules: SAM requires a standalone model for representation abstraction and a model for generating embeddings; MAP only needs a memory bank for storing historical embeddings and computing penalties. Neither is tied to the R-Zero family.
>
> To provide initial evidence that R-Diverse can transfer to more general domains, we built a general-purpose SAM variant, replacing the Coder with Qwen2.5-7B-Instruct (same 7B budget) to paraphrase questions into structured text, and using Qwen3-Embed-0.6B instead of a code-specific embedder. This variant only drops 1.05 pts from the original SAM while still outperforming w/o SAM by +1.04 (see W2 in our response to Reviewer t4Jx for details).
>
> ---
> ### **W5: Code abstraction justification**
>
> Providing a theoretical justification that code is the uniquely correct abstraction is admittedly difficult. To empirically investigate this, we conducted a further ablation: replacing code generation in SAM with paraphrasing while keeping the embedding model unchanged. This results in a 1.37-point drop (52.59 to 51.22), indirectly reflecting the advantage of code as an abstraction method.
>
> ---
> ### **Key Q2: Sensitivity to base model, coder, and embedding model**
>
> We conducted sensitivity analyses varying the base model, coder, and embedding model. see W1 & Key Q1 in our response to Reviewer t4Jx for the full table.
>
> ---
> ### **Key Q3: Failure cases of R-Diverse**
>
> We acknowledge that in our early design, the memory bank did over-penalize productive revisiting of earlier skill regions. To address this, we adopted two measures: (1) a threshold mechanism (tau_max=0.5, tau_mean=0.25) to control penalty strength, and (2) memory replay to ensure the Solver retains access to earlier questions. However, we have not yet systematically characterized incorrect-abstraction failures where genuinely different problems are treated as similar, which remains an important limitation driving future work. We also observed an interesting phenomenon (see [Figure 2](https://anonymous.4open.science/r/R8E6/) in https://anonymous.4open.science/r/R8E6/): when R-Diverse converges, Cross- and Intra-Iteration Repetition metrics converge simultaneously, **offering a diversity-centric explanation for why self-play systems cannot sustain indefinite improvement**.
>
> ---
> ### **Presentation**
>
> We will fix the noted typos and make the R-Diverse* fairness comparison more explicit in revision.
>
> ---
> Once again, we sincerely appreciate the time and expertise you have invested in reviewing our work!
>
> Best,
> Authors

---

> > ### Author Rebuttal · Reviewer_kmEj · 2026-04-02
> >
> > Thank you for the thorough reply to my review and engagement with my questions and concerns. I still see the main limitation as **evidence of generalization to other self-play frameworks** and an **analysis of failure cases**. Additional evidence for MAP and SAM transferring to self-play frameworks would help improve the significance of the work. Analysis of failure cases would help provide a better understanding of the overall method.

---

> > > ### Author Response · Authors · 2026-04-04
> > >
> > > Dear Reviewer kmEj,
> > >
> > > Thank you for your continued engagement and for recognizing the effort in our previous response! Your suggestion to include cross-framework evidence and failure analysis is entirely on point and **significantly strengthens** our work. Below, we provide **concrete empirical evidence** addressing your remaining concerns, alongside a **pleasant surprise** regarding your "finer-grained penalty" suggestion.
> > >
> > > ---
> > > ### **1. Generalization to Other Self-Play Frameworks**
> > > While R-Zero is a foundational paradigm spawning many methods (e.g., Agent-0, VisPlay, Active-Zero), we completely agree that demonstrating broader transferability is crucial. To prove this, we integrated MAP and SAM into two representative frameworks:
> > > - **Absolute Zero (`Out-of-R-Zero-family`):** A continuous, single-model online training paradigm, fundamentally different from R-Zero's dual-model iterative loop:
> > > | Method | Step | Math AVG |
> > > |:-|:-:|:-:|
> > > | ***Ref. (Qwen3-4B-Base)*** | | |
> > > | Absolute Zero | 350 | 46.42 |
> > > | Absolute Zero | 500 | 44.95 |
> > > | ***Main Results (vs. Ref)*** | | |
> > > | Absolute Zero + MAP + SAM | 350 | 47.56 (+1.14) |
> > > | Absolute Zero + MAP + SAM | 500 | 48.47 (+3.52) |
> > > | ***Ablation (vs. Main)*** | | |
> > > | Absolute Zero + MAP | 500 | 47.71 (−0.76) |
> > > | Absolute Zero + SAM | 500 | 47.22 (−1.25) |
> > > - **VisPlay (Inner-R-Zero-family, `Cross-modality`, `General task`):** An extension of R-Zero to VLMs handling more general tasks (so using our general-purpose SAM variant here):
> > > | Method | Iteration | AVG |
> > > |:-|:-:|:-:|
> > > | ***Ref. (Qwen2.5-VL-7B-Instruct)*** | | |
> > > | VisPlay | 3 | 48.55 |
> > > | VisPlay | 5 | 46.01 |
> > > | ***Main Results (vs. Ref)*** | | |
> > > | VisPlay + MAP + SAM (gen) | 3 | 49.41 (+0.86) |
> > > | VisPlay + MAP + SAM (gen) | 5 | 50.34 (+4.33) |
> > > | ***Ablation (vs. Main)*** | | |
> > > | VisPlay + MAP | 5 | 49.82 (−0.52) |
> > > | VisPlay + SAM (gen) | 5 | 48.77 (−1.57) |
> > >
> > > As shown above, the baseline results confirm that `late-stage performance degradation is a systemic issue across different frameworks`. By introducing MAP and SAM, R-Diverse successfully `mitigates this issue in both, unlocking sustained improvements with clear ablation benefits.`
> > >
> > > ---
> > > ### **2. Deep Dive into Failure Cases**
> > > To check if SAM incorrectly treats genuinely different problems as similar, we checked **all 21,424 questions** used for training across 5 iterations via GPT-5.4 screening and manual review. In the vast majority of cases, SAM works exactly as intended. However, the semantic abstraction degrades in two specific modes (**see Table 3 in our repo** https://anonymous.4open.science/r/R8E6/ for specific examples):
> > > - **Code Generation Collapse (~0.7%):** For highly complex questions, the Coder occasionally fails, defaulting to a trivial *def solver(): return 1* as a blind guess when unable to solve the problem. This indeed inflates similarity between genuinely different problems. However, given its negligible frequency, it doesn't undermine R-Diverse's gains and primarily stems from the Coder's capacity limits rather than SAM's mechanism.
> > > - **Verbosity in Structured Math (~7%):** For structured data (e.g., matrices, graphs), the Coder generates verbose, low-level code instead of abstract logic, diluting the core semantic information.
> > >   - *Fix & Improvement:* We adjusted the Coder's prompt to enforce the use of standard scientific libraries (e.g., numpy, networkx) when encountering structured data. This simple fix further `boosted R-Diverse's performance from 52.59 to 53.02 (+0.43)`, perfectly validating your suggestion to investigate failure cases!
> > >
> > > More fundamentally, as illustrated in **Figure 2 in our repo**, the **eventual performance plateau stems from the fact that question diversity cannot scale infinitely**. This phenomenon beautifully `validates the tight correlation between diversity and performance`, and even `highlights a promising direction for further unlocking the sustained evolutionary potential` of self-play systems.
> > >
> > > ---
> > > ### **3. Finer-Grained Penalty (Your Suggestion)**
> > > We explored your excellent suggestion of replacing the coarse global mean with two finer-grained penalties:
> > > - **Density-Aware Penalty (KNN):** Penalizes the average similarity to the $K$ nearest historical neighbors.
> > > - **Cluster-Aware Penalty (K-Means):** Penalizes the sum of similarities to cluster centroids, weighted by cluster size.
> > >
> > > We have **included the results in Table 4 of our repo**, which demonstrate that your intuition was remarkably accurate! While the Cluster-Aware approach (acting somewhat similarly to the global mean) provided stable but modest gains, the Density-Aware approach `yielded a substantial +0.79 improvement`, demonstrating `the great potential of finer-grained penalty`.
> > >
> > > ---
> > > We sincerely thank you for these valuable suggestions and hope this empirical evidence fully resolves your remaining questions. If accepted, all new findings will be seamlessly incorporated into the Camera-Ready version without major manuscript updates.
> > >
> > > Best, Authors

---

### Official Review · Reviewer_t4Jx · 2026-03-23

**Soundness:** 3
**Presentation:** 3
**Significance:** 3
**Originality:** 2
**Overall Recommendation:** 4
**Confidence:** 2

**Summary:**

This paper proposes two improvements to self-play LLM training: “Memory-Augmented Penalty (MAP), which uses a persistent memory bank to discourage recycling across iterations, and Skill-Aware Measurement (SAM), which evaluates diversity by the reasoning skills exercised rather than surface variation of questions.”

**Compliance With Llm Reviewing Policy:**

Affirmed.

**Final Justification:**

I thank the authors for their response and have increased my overall score from 3 to 4.

**Key Questions For Authors:**

1.	Why Jina-Code-Embeddings-1.5B?
2.	“R-Diverse demonstrates sustained, monotonic improvement over five evolution”. Why stop at 5? When does perf decrease / converge?

**Strengths And Weaknesses:**

-> Strengths:

1.	Authors propose R-Diverse with two intuitive innovations: Memory-Augmented Penalty (MAP), which uses a persistent memory bank to discourage recycling across iterations, and Skill-Aware Measurement (SAM), which evaluates diversity based on the reasoning skills exercised rather than the surface variation of questions.
2.	Across 10 math and general reasoning benchmarks, R- Diverse sustains gains over more iterations and consistently outperforms prior self-play methods.



-> Weaknesses:

1.	How generalizable are the methods across base models (only qwen3 family is used) and other LLMs used in the intermediate steps (Qwen2.5-Coder-7B for solution gen, Jina-Code-Embeddings- 1.5B)?
2.	R-Diverse has a “representation abstraction step that maps each question to a canonical solver-level program representing its solution procedure”. How generalizable is this step across domains? How accurate is this LLM-prompting when evaluated by human annotators?
3.	“Despite this promise, current self- play frameworks for reasoning often yield non-sustained gains: performance improves early but plateaus or degrades after a few iterations”. Can a thorough analysis be provided: For each baseline and R-Diverse, what’s the accuracy vs iteration trends per dataset, not overall average? When does R-Diverse converge and/or decrease? The overall average hides individual dataset performance since datasets have widely different sample sizes.
4.	How were the hyperparameters chosen? 5 training steps for challenger, 15 for solver, etc? How robust is the method to parameter choices?
5.	Can the result trends be explained: domain or dataset-wise or model-wise? Why is R-Diverse behind in general reasoning (SuperGPQA)? For the 8B model, R-Diverse is behind on some math benchmarks? Any analysis as to why? Overall general reasoning average is not included in Table 1, only overall math average.

---

> ### Author Rebuttal · Authors · 2026-03-30
>
> Dear Reviewer t4Jx,
>
> Thank you for your constructive feedback! We are encouraged by your recognition that our innovations (MAP and SAM) are `intuitive` and that R-Diverse `sustains gains over more iterations and consistently outperforms prior self-play methods`. Below, we provide point-by-point responses. Hope our response could address your concerns!
>
> ---
> ### **W1 & Key Q1: Generalizability across base models, coder/embedding choices & Why Jina-Code?**
>
> (a) Base model family. Following R-Zero's cross-family setting, we evaluated on OctoThinker 3B & 8B for 5 iterations (see Figure 1 in our anonymous repo: https://anonymous.4open.science/r/R8E6/). R-Diverse consistently outperforms R-Zero on both scales (+1.58 / +1.55 at peak). The 3B peaks at iter 4 while 8B sustains improvement up to iter 5 (consistent with R-Zero's finding that self-play capacity scales with base model size), demonstrating R-Diverse's cross-family generalization.
>
> (b) SAM component sensitivity. We tested 2 alternative coders and 3 alternative embeddings (see W1 & Q3 in our response to Reviewer ic2V for the full table). All variants outperform w/o SAM (+1.24 to +2.09), confirming SAM's benefit is not tied to a specific model.
>
> ---
> ### **W2: Generalizability and accuracy of representation abstraction**
>
> (a) Domain generalizability. As noted in our Conclusion, the code-based implementation is tailored to math, but SAM's core idea — measuring diversity by skills exercised — is domain-agnostic. We provide a general-purpose variant: replacing the Coder with Qwen2.5-7B-Instruct (same 7B budget) for paraphrasing, and using Qwen3-Embed-0.6B:
>
> ||||
> |-|:-:|:-:|
> |Setting|Math AVG|Δ vs w/o SAM|
> |Full SAM (code + code-embed)|52.59|+2.09|
> |General SAM (paraphrase + general-embed)|51.54|+1.04|
> ||||
>
> This variant needs no code-specific models, balancing generality and performance, and enables SAM to extend to broader domains.
>
> (b) Code generation accuracy. We randomly sampled 200 examples for human evaluation, finding ~68% accuracy (73% by GPT-5.4 on 10x samples). However, exact correctness is not critical: code serves as a semantic bottleneck to reflect underlying skills. Even imperfect code preserves dependency structures and operation patterns, which is why SAM improves across all configurations.
>
> ---
> ### **W3 & Key Q2: Per-dataset iteration trends & convergence**
>
> (a) Per-dataset results. Other baselines (Absolute Zero, SPIRAL, Socratic-Zero) use continuous online training or external-teacher-based evolution, lacking a comparable "iteration" idea (as detailed in our reply to Reviewer kmEj, W2). We provide per-bench results for R-Zero and R-Diverse in Table 1 (anonymous repo).
> It's worth noting that, Appendix Tables 4-5 have already reported the detailed per-dataset results for R-Diverse, and all AVGs presented in our paper are averaged within each bench first then across benches, avoiding sample-size bias.
>
> (b) Extended training dynamics. We trained R-Diverse up to 8 iterations on Qwen3:
>
> ||||||
> |-|:-:|:-:|:-:|:-:|
> |iter|5|6|7|8|
> |4B|52.59|52.74 (peak)|52.61|52.47|
> |8B|56.46|56.73|56.77 (peak)|56.49|
> ||||||
>
> Gains beyond iter 5 are marginal. As shown in Figure 2 (anonymous repo), Cross- and Intra-Iteration Repetition converge simultaneously with performance, consistent with the diversity-bottleneck hypothesis.
>
> ---
> ### **W4: Hyperparameter choices & robustness**
>
> All hyperparameters except those from MAP/SAM are inherited from R-Zero for fair comparison. For key parameters, sensitivity analyses (Table 2, anonymous repo) show a +0.24 improvement when increasing solver training steps to 20, indicating that R-Diverse still has room for further gains with tuning.
>
> ---
> ### **W5: Result trends & benchmark performance**
>
> (a) Trend explanation. Gains are strongest on math benchmarks where our curriculum directly applies, and smaller on knowledge-intensive general benchmarks where transfer is indirect, e.g. SuperGPQA.
>
> (b) Benchmark-specific clarification. R-Diverse (3/5 iter) actually achieves SOTA on all three general benchmarks including SuperGPQA. On Olympiad/AIME, the only method ahead is Socratic-Zero, which leverages a much larger model (Qwen3-235B-A22B) for curriculum generation, while R-Diverse self-evolves using only small 4/8B models. Excluding Socratic-Zero, R-Diverse is SOTA on Olympiad and AIME24; the AIME25 gap likely reflects self-play variance on small-sample (30-question) evaluations.
>
> (c) General reasoning average. Overall AVG is included in Table 1. We also provide a separate General AVG:
>
> ||||
> |-|:-:|:-:|
> |Scale|iter3|iter5|
> |4B|31.51 (SOTA) |31.38|
> |8B|35.32|35.51 (SOTA)|
> ||||
>
> ---
> Once again, we sincerely appreciate your time and valuable feedback. We hope the above responses and new experiments fully address your concerns!
>
> Best,
> Authors

---

> > ### Author Rebuttal · Reviewer_t4Jx · 2026-04-04
> >
> > I thank the authors for their response and have increased my overall score from 3 to 4.

---

> > > ### Author Response · Authors · 2026-04-07
> > >
> > > Dear Reviewer t4Jx,
> > >
> > > Thank you so much for reviewing our rebuttal and for increasing your overall score to 4! We deeply appreciate your recognition of our work. We noticed that the system status is marked as *"Partially resolved - I have follow-up questions for the authors."* However, we didn't find any specific questions in the comment text. **If you have any remaining concerns or need further clarifications, please feel free to share them with us!** We are **standing by** and would be more than happy to address them! :-D
> > >
> > > In the meantime, while addressing other reviewers' questions, we obtained some exciting new results over the past few days. Given your initial interest in **R-Diverse's generalizability (W1, W2), representation abstraction (W2)**, we believe these new findings will be highly relevant:
> > >
> > > ### **1. R-Diverse Shows Comprehensive Generalization (Across Base Models, Frameworks, and Tasks)**
> > >
> > > To demonstrate that the robustness of R-Diverse goes well beyond the choice of SAM components and base models, we further evaluated its broad generalizability as a plug-and-play method **across frameworks and tasks.** Specifically, to prove that R-Diverse is not limited to the framework from R-Zero family and mathematical tasks, we integrated MAP and SAM into two fundamentally different frameworks:
> > >
> > > - **Absolute Zero** (**`Out-of-R-Zero-family`**, **Coding task**): A continuous, single-model online training paradigm, fundamentally different from R-Zero's dual-model iterative loop.
> > > - **VisPlay** (**Inner-R-Zero-family, `Cross-modality`, `General task`**): An extension of R-Zero to VLMs handling more general tasks (so using our general-purpose SAM variant here).
> > >
> > > As detailed in **Tables 5 & 6 in our repo (https://anonymous.4open.science/r/R8E6/)**, the baseline results confirm that `late-stage performance degradation is a systemic issue across different frameworks and tasks`. By introducing MAP and SAM, `R-Diverse successfully mitigates this issue in both, unlocking sustained improvements with clear ablation benefits`:
> > >
> > > - **Absolute Zero + R-Diverse**: Math AVG at step 500 improved from 44.95 to **48.47 (+3.52)**.
> > > - **VisPlay + R-Diverse (general-purpose variant)**: General task AVG at iteration 5 improved from 46.01 to **50.34 (+4.33)**.
> > >
> > > These results confirm that `R-Diverse functions as a robust plug-and-play method`, whose effectiveness is `not only insensitive to the choice of SAM's components and base models (as addressed in our previous rebuttal) but also generalizes across diverse self-play frameworks and task domains`.
> > >
> > > ### **2. R-Diverse Failure Case Analysis & Immediate Fix**
> > >
> > > Prompted by your concern about the representation abstraction step, we investigated further and found that while SAM functions exactly as intended in the vast majority of cases, there are indeed two edge-case failure modes (**see Table 3 in our repo for specific examples**):
> > >
> > > - **Code Generation Collapse (~0.7%)**: For highly complex questions, the Coder occasionally fails, defaulting to a trivial *def solver(): return 1* as a blind guess. However, given its negligible frequency, it doesn't undermine R-Diverse's gains.
> > > - **Verbosity in Structured Math (~7%)**: For structured data (e.g., matrix, graph), the Coder generates verbose, low-level code instead of abstract logic. **The Fix:** We simply adjusted the Coder's prompt to enforce the use of standard scientific libraries (*numpy*, *networkx*) when encountering structured data, which `further boosts R-Diverse's performance from 52.59 to 53.02 (+0.43)`.
> > >
> > > ### **3. R-Diverse Actively Preserves Question Quality**
> > >
> > > Beyond evaluating the code, we also analyzed the quality of the questions generated for training. Specifically, we employed GPT-5.4 as an independent judge to evaluate questions' quality across 5 iterations. Surprisingly, **74.3%** of the questions generated by R-Diverse were judged to be logically sound and free of contradictions, compared to **only 65.7%** for the R-Zero baseline.
> > >
> > > This reveals a hidden strength of SAM: `it acts as a semantic bottleneck`. While surface metrics (e.g., BLEU) can be **hacked** by the Challenger generating random noise (which easily yields high textual variance), `random noise fails SAM's abstraction step and receives zero reward`. Thus, `SAM naturally regularizes against the "noise-for-diversity" hacking`.
> > >
> > > ### **4.  R-Diverse Achieves Higher Peaks via Finer-Grained Penalty**
> > >
> > > Inspired by Reviewer kmEj, we explored replacing the coarse global-mean penalty with finer-grained alternatives: a **Density-Aware Penalty** (KNN-based) and a **Cluster-Aware Penalty** (K-Means-based). The former yielded a substantial improvement (**see Table 4 in our repo for details**), pushing R-Diverse's performance from 52.59 to **53.38 (+0.79)**.
> > >
> > > We will incorporate all these new analyses into the final version of the paper. And we look forward to any follow-up questions you might have! :-D
> > >
> > > Best, Authors

---

### Decision · Program_Chairs · 2026-04-30

**Decision:**

Accept (regular)

**Comment:**

This paper received a borderline score, with some reviewers are not fully convinced by the rebuttal, although all the reviewers have acknowledged the rebuttal, but they have not been as engaged as much as other reviewers. The AC has went through the reviews, but it is hard to further evaluate when the reviewer didn't specify the remaining concerns. The AC has also initiated an internal discussion but didn't change much of the current evaluation. There is also an author-AC confidential post regarding the not-perfectly-responsiveness of the reviewers.

Overall, even the AC have decided to slightly penalize the reviewers that are not-so-perfectly responsive, leading to a very borderline situation: acceptable if there is room in the program.